# Multi-Response Optimization of Surface Grinding Process Parameters of AISI 4140 Alloy Steel Using Response Surface Methodology and Desirability Function under Dry and Wet Conditions

Rakesh Roy [1], Sourav Kumar Ghosh [2], Tanvir Ibna Kaisar [3], Tazim Ahmed [1], Shakhawat Hossain [1], Muhammad Aslam [4], Mosab Kaseem [5,*] and Md Mahfuzur Rahman [1,*]

1   Department of Industrial and Production Engineering, Jashore University of Science and Technology, Jashore 7408, Bangladesh; rakeshroy996@gmail.com (R.R.); tazim_ipe@just.edu.bd (T.A.); shakhawat.ipe@just.edu.bd (S.H.)
2   Department of Industrial and Production Engineering, Bangladesh University of Textile, Dhaka 1208, Bangladesh; sourav@butex.edu.bd
3   Department of Industrial and Production Engineering, Bangladesh University of Engineering and Technology, Dhaka 1000, Bangladesh; tanvirkaisar@ipe.buet.ac.bd
4   Department of Chemical Engineering, COMSATS University Islamabad, Lahore 54000, Pakistan; maslam@cuilahore.edu.pk
5   Department of Nanotechnology and Advanced Materials Engineering, Sejong University, Seoul 05006, Korea
*   Correspondence: mosabkaseem@sejong.ac.kr (M.K.); mrahman.ipe@just.edu.bd (M.M.R.)

**Abstract:** The effect of four controllable input process parameters of AISI 4140 steel, cross-feed, workpiece velocity, wheel velocity, and the depth of cut were experimentally investigated under dry and wet conditions. Three responses, contact temperature, material removal rate (*MRR*), and machining cost during surface grinding of AISI 4140 steel, were considered. The process was optimized using a recently developed combined methodology based on response surface methodology (RSM) and desirability functional approach (DFA). RSM generated the models of the responses for prediction while DFA solved these multi-response optimization problems. The DFA approach employed an objective function known as the desirability function, which converts an estimated response into a scale-free value known as desirability. The optimum parameter was attained at the maximum overall desirability. An analysis of variance (ANOVA) was conducted to confirm the model adequacy. From the results of the study, for equal weights of responses, the corresponding optimal values of the input parameters cross-feed, workpiece velocity, the wheel or cutting velocity and the depth of cut were found to be 6 mm/pass, 12 m/min, 15 m/s, and 0.095 mm respectively in wet conditions. The corresponding predicted output responses were: 134.55 °C for the temperature, and 7.366 BDT (Taka, Currency of Bangladesh) for the total cost with an overall desirability of 0.844. Confirmation testing of optimized parameters, i.e., checking the validity of optimal set of predicted responses with the real experimental run were conducted, and it was found that the experimental value for temperature and total cost were 140.854 °C and 8.36 BDT, respectively, with an overall desirability of 0.863. Errors of the predicted value from the experimental value for equal weightage scheme were 4.47% for the temperature and 7.37% for the total cost. It was also found that if the temperature was prioritized, then the wet condition dominated the overall desirability, which was expected. However, if the cost was given high weightage, dry condition achieved the highest overall desirability. This can be attributed to the cutting in the wet condition which was more expensive due to the application of cutting fluid. The proposed model was found to be new and highly flexible in the sense that there was always an option at hand to focus on a particular response if needed.

**Keywords:** RSM; DFA; dry grinding; wet grinding; surface grinding

## 1. Introduction

Grinding is widely used as an ultimate machining process in manufacturing of components incorporating excellent geometric accuracy and satisfactory surface finish. Compared with other machining processes, such as turning and milling, grinding process generates a substantial amount of heat at the grinding zone (almost 10–30 times of that in turning) due to the very high energy input per unit volume of material removal raising the temperature of the grinding zone [1,2]. Excessive temperature may cause thermal damage to the workpiece such as workpiece burn, phase transformations, undesirable residual tensile stresses, cracks, reduced fatigue strength, and thermal distortion and inaccuracies [3]. It may also cause premature failure of the cutting tools, rapid oxidation, and corrosion [4,5]. The most commonly used dry grinding has the advantages of cost-effectiveness and environment friendly nature with no pollution despite having major drawbacks namely thermal damage, high friction, high residual stress, and high wheel-wear [6]. On the other hand, wet cutting condition removes heat from the cutting zone. However, it does not always seem to be effective as it consumes a large amount (i.e., 8 L/min) of coolant which is not economically feasible and has a significant impact on the ecology and internal environment (workshop) [7,8]. Hence, a trade-off among material removal rate (*MRR*), cutting zone temperature, and machining cost is needed for effective machining.

The magnitude of residual stresses caused by grinding is determined by the physical and mechanical characteristics of the grinding material as well as the parameters, such as wheel speed, workpiece speed, depth of cut, grinding wheel composition, and lubrication. These, in turn, determine the temperature gradient in the grinding zone [9]. In steels, an empirical connection between peak residual tensile stress and maximum grinding zone temperature has been identified [10].

Investigating cutting zone temperature and heat transfer in grinding requires specific temperature measurements. Thermal imaging, optical fiber, foil/workpiece (single pole) thermocouple, and embedded (double-pole) thermocouple are some prominent temperature measuring techniques. Because of its relative simplicity, low cost, precision, and dependability, the integrated thermocouple technique is the most commonly utilized. A two pole thermocouple was welded to the bottom of a blind hole drilled close to the ground surface from the underside of the workpiece using this approach [11,12]. Changfeng Yao et al. investigated the impact of wheel speed and depth of cut on the temperature of Aermet 100 steel during surface grinding using three different types of wheels. As a temperature measurement device, they employed a single alumina wheel, a white alumina wheel, and a CBN wheel with coolant. A thermocouple was also used. They discovered that the overall heat flow of a single alumina wheel is the greatest, which is due to two factors: high tangential grinding force and poor thermal properties of the grinding wheel [13].

The response surface methodology (RSM) is a collection of mathematical and statistical approaches that may be used to model and analyze issues in which a response of interest is influenced by multiple factors and the goal is to maximize this response [14]. Bhushan used the response surface method and desirability analysis to examine the impact of cutting parameters during the turning of 7075 Al alloy SiC composite in order to minimize machine power consumption and enhance tool life [15]. Using RSM technique, Rudrapati et al. investigated the influence of cylindrical grinding process parameters on responses such as workpiece vibration and surface roughness [16]. A multi-objective genetic algorithm was used to optimize process parameters for the required results, and the projected model was validated using a confirmatory test. M. Janardhan and A. Gopala Krishna used response surface methodology (RSM) to determine the optimum machining parameters leading to minimum surface roughness and maximum metal removal rate in surface grinding process operation on EN24 steel [17]. Aravindand Periyasamy optimized process parameters by using the Taguchi method and response surface methodology (RSM) for surface grinding of AISI 1035 steel and they considered grinding wheel abrasive grain size, depth of cut, and feed as process parameters and Surface roughness as output response [18].Venkatesan et al. focused on the comparative performance of input parameters viz. speed, feed, and

depth of cut on cutting force, roughness, tool wear, and chip morphology of dry and MQL machining conditions on Nimonic 90 alloy with the aid of RSM [19]. Manohar et al. optimized output responses viz. surface roughness, cutting forces, and *MRR* as a function of input process parameters, viz. cutting speed, feed rate, and depth of cut with RSM and desirability function approach (DFA) while turning Inconel 718 using coated carbide tools [20]. Naresh et al. focused on the optimization of process parameters namely, cutting speed, feed rate, and depth of cut which influence the output response surface roughness and material removal rate with grey relational analysis (GRA) and desirability function analysis (DFA) in CNC milling of AISI 304 stainless steel [21]. Buranska et al. demonstrated multi-criterial optimization with desirability function analysis (DFA) of input factors cutting environment, feed for two defined target functions roughness, and cylindricity for drilling of aluminum alloys [22]. Borchers et al. investigated different manufacturing processes of AISI 4140 steel with regard to surface modification but not identified the optimum machining parameters [23]. However, unfortunately, very little literature is available on process parameter optimization of surface grinding with DFA. A comprehensive summary from previous literature on grinding type, the material used, optimized input parameters, and output responses along with the optimization (analysis) techniques of surface grinding is presented in Table 1.

It is apparent from the literature that the process parameters such as cross-feed, workpiece velocity, wheel velocity, and the depth of cut have a significant influence on the temperature, *MRR*, and production cost in the grinding process for different cutting conditions. The temperature in turn affects the finishing of the workpiece and tool life. To the best of the authors' knowledge, no study has been conducted on the surface grinding of AISI 4140 steel to evaluate, compare, and optimize contact temperature, *MRR*, and machining cost in dry and wet machining conditions so far. Again, the combined methodology of RSM and DFA in optimization of machining parameter is rarely found in this relevant field. Combined approach of RSM and DFA known as desirability optimization methodology (DOM) is efficient in determining the optimum antagonist responses (here temperature and *MRR*) which are led by different input machining parameter.

This study looks into the grindability of AISI 4140, a chromium-molybdenum alloy steel. The chromium component allows for effective hardness penetration, while the molybdenum element guarantees consistent hardness and strength. Carbon steel is alloyed with one or more alloying elements such as manganese, silicon, nickel, titanium, copper, chromium, and aluminum to create alloy steels. These metals are added to provide characteristics not seen in standard carbon steel. Superior toughness, ductility, and wear resistance under quenched and tempered conditions are among the desired characteristics of AISI 4140 [23]. Hence it has a lot of applications in the engineering field.

Thus, in this research work, we optimize contact temperature, *MRR*, and machining cost during surface grinding of AISI 4140 steel using a combined methodology based on surface methodology (RSM) and desirability functional approach (DFA). From the literature study presented above and in the Table 1, it can be said that the optimization of surface grinding process parameters of AISI 4140 steel using a combined methodology based on surface methodology (RSM) and desirability functional approach (DFA) for both wet and dry condition is rarely studied. Thus, the novelty of this work is to study the multi-objective optimization with response surface methodology (RSM) and desirability functional approach (DFA) of surface grinding of AISI 4140 steel. Therefore, the contribution of this research can be outlined as follows:

1. Experimentally investigation the influence of the input machining parameters (cross-feed, workpiece velocity, wheel velocity, and the depth of cut) on three responses (contact temperature, material removal rate (*MRR*), and machining cost) during surface grinding of AISI 4140 steel.
2. Establishment of an empirical equation of each output responses in terms of input parameters based on the experimental data. Further this mathematical model is used to predict the output responses.

3.  Optimization of the output responses (minimum contact temperature, maximum *MRR*, and minimum machining cost) based on the DFA methodology. Further, optimal set of input parameters are investigated for these optimal responses.

**Table 1.** Grinding process parameters optimization using different analysis techniques in the literature.

| Type of Grinding | Material Used | Parameter Optimized | Optimized Output | Analysis Technique | Ref. |
|---|---|---|---|---|---|
| Dry grinding | Mild steel | Wheel speed, workpiece speed, depth of dressing and lead of dressing | Minimum production cost, maximum production rate and the finest possible surface grinding finish | Genetic Algorithm (GA) | [24] |
| | Mild Steel | Wheel speed, workpiece speed, depth of dressing and lead of dressing | Production cost, production rate and surface finish | Particle swarm optimization (PSO) algorithm | [25] |
| | Mild Steel | Wheel speed, workpiece speed, depth of dressing and lead of dressing | Production cost, production rate and surface finish | Quantum Based Optimization Method (QBOM) | [26] |
| | Mild Steel | Wheel speed, workpiece speed, depth of dressing and lead of dressing | Production cost, production rate and surface roughness | Hybrid Particle Swarm Optimization (HPSO) algorithm | [27] |
| | 1.2080 Steel | Wheel speed, workpiece speed and depth of cut | Surface finish, total grinding time and production cost | Non-dominated sorting genetic algorithm (NSGA II) | [28] |
| | 1.2080 Steel | Wheel speed, workpiece speed and depth of cut | Surface quality, total grinding time and production cost | Dragonfly algorithm | [29] |
| | EN24 steel | Wheel speed, table speed and depth of cut | Surface roughness and metal removal rate | RSM optimization | [17] |
| | Soda-lime glass | Wheel speed, depth of cut and feed rate | Surface roughness | RSM and Monte Carlo Simulation | [30] |
| | Steel | Speed of wheel, speed of workpiece, depth of dressing and lead of dressing | Production cost, production rate and surface finish | Particle swarm optimization, Gravitational search algorithm and Sine Cosine algorithm | [31] |
| | Stainless steel material AISI 304 | Feed rate, speed of table and depth of cut | *MRR* and surface roughness | ANOVA, Taguchi method | [32] |
| | EN 24 steel | Wheel speed, depth of cut and feed rate timing | Surface roughness | RSM | [33] |
| Conventional grinding | Silicon nitride ceramic | Feed rate, depth of cut, type of diamond wheel and lubrication type | Grinding forces, workpiece surfaceroughness, surface damages and wheel wear | Adaptiveneuro-fuzzy inference system (ANFIS) and Taguchi method | [34] |
| | Tungsten carbide insert | Feed rate and cutting speed | Production costs, grinding burn, surface roughness and temperature at the grinding surface | Constrained Bayesian optimization combined with Gaussian process Models | [35] |
| | Ti-6Al-4V | Coolant types, cooling techniques and grinding depths | Surface hardness and surface morphology | Taguchi method and ANOVA | [36] |
| | 9CrSi annealing tool steel | Coolant concentration, coolant flow, cross-feed, table speed and depth of cut | Surface roughness | Taguchi method | [37] |
| | Ti-6Al-4V-ELI | Types of coolant and graphene percentage in the coolant | Surface roughness, grinding force, specific grinding energy and coefficient-of-friction | Experimental (conventional) | [38] |

**Table 1.** *Cont.*

| Type of Grinding | Material Used | Parameter Optimized | Optimized Output | Analysis Technique | Ref. |
|---|---|---|---|---|---|
| MQL grinding | EN8 flat plate | Depth of cut, type of lubricant, feed rate, grinding wheel speed, coolant flow rate and nanoparticlesize | G ratio and surface finish | Taguchi based Grey relational analysis, ANOVA | [39] |
| | Two soft steels (CK45 and S305) and two hard steels (HSS and 100Cr6) | Depth of cut, cutting speed and feed rate | Grinding forces, friction coefficient, surface roughness, surface morphology and form of the chips | RSM, ANOVA, GA | [40] |
| Electrochemical grinding | 100Cr6 hardened steel (bearing steel) | Specific material removal rate | Maximum temperature, grinding forces and friction coefficient | Experimental (conventional) | [41] |
| | Composite carbide inserts | Voltage and cutting speed | Current density, material removal rate (*MRR*) and surface finish | Response surface methodology (RSM), Desirability function | [42] |

Finally, DFA aids to find out optimal set of input parameters along with their corresponding output responses such as contact temperature, *MRR*, and machining cost. In addition, DFA further optimizes the input process parameters to get the minimum contact temperature, maximum *MRR*, and minimum machining cost.

## 2. Methodology and Experimental Procedures

The materials and methods have been described with sufficient details to allow others to replicate and build on the published results.

In this study, RSM and DFA have been anticipated as solution methodology. The outline of this research is summarized in Figure 1. Materials and instruments, experimental setup, RSM, and DFA methods are detailed in the subsequent subsections.

### 2.1. Materials and Instruments

Workpiece: The surface grinding operation is performed on AISI 4140 alloy steel (Bozhong Metal Group, Shanghai, China). It is a chromium-molybdenum alloy steel. The chemical composition, dimension, and other physical properties of the workpiece material are listed in Table 2.

Grinding wheel: Grinding wheel is the basic cutting tool in grinding operation. The grinding wheel used in this experiment was alumina oxide vitrified, i.e., glassy and non-crystalline, from NORTON (Shanghai, China). Its dimension is 14 in × 1.5 in × 3 in. This grinding wheel (FE 38A100L5V) is used in accordance with the machining circumstances and manufacture's exhortations. According to the standard system of grinding wheel markings, the specifications of the grinding wheel are: type of abrasive: aluminum oxide; grit size or the grain size: 100 μm, i.e., fine; grade: L, indicating medium, i.e., neither soft nor hard; the porosity or the dense to open: 5; and the type of bonding: V, i.e., vitrified. A Single point diamond, HS050 (Shanghai Sanxin Diamond Tools Co., Ltd., Shanghai, China) dresser was used to dress the grinding wheel. A total depth of dressing 40 μm was achieved.

Cutting fluid: This experiment was carried out both in dry and wet conditions. Castrol Syntilo 9954 (Shangai, China) was selected as cutting fluid in wet machining conditions. An emulsion was made on this synthetic coolant basis with concentration of 5%. Its density is 1066 kg/m$^3$. Single nozzle open metal-tube type was used to apply coolant at a flow rate 5 L/min. Flow rate was measured with Z-5615 Panel Flowmeter (Emerson Electric Limited, Rayong, Thailand).

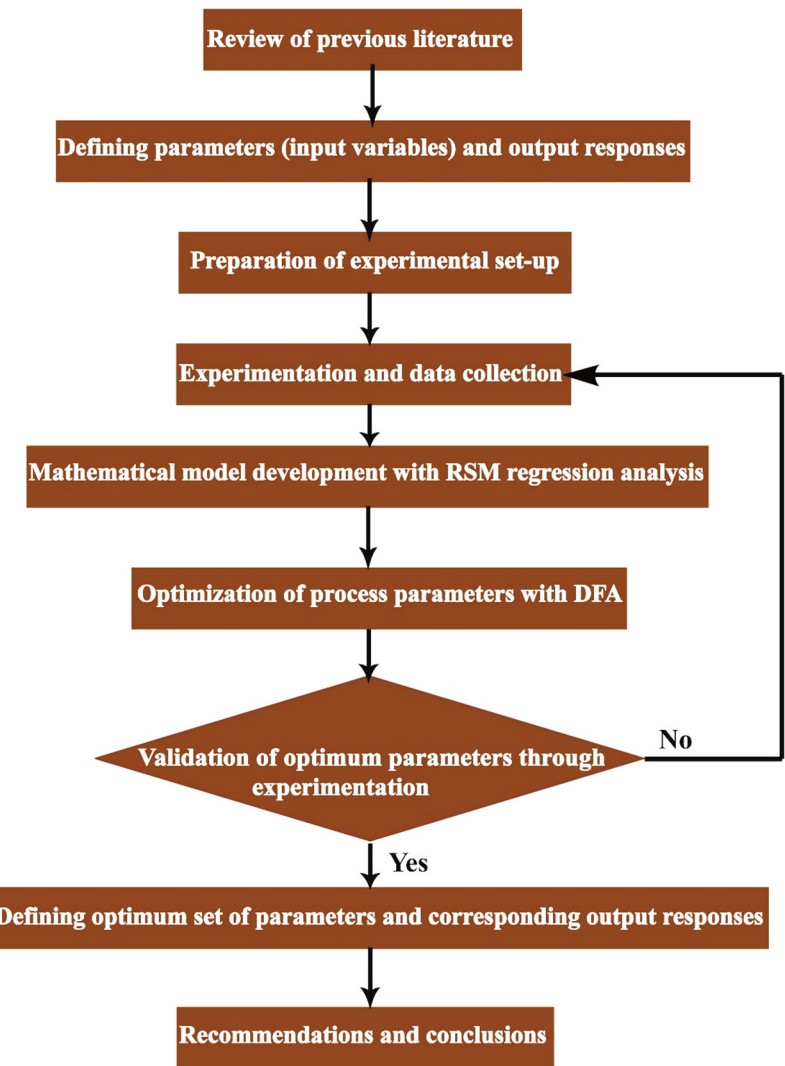

**Figure 1.** Flow chart of current research.

**Table 2.** Chemical composition and properties of the workpiece material.

| Properties | Specification | | | | | | | |
|---|---|---|---|---|---|---|---|---|
| Work material | AISI 4140 alloy steel | | | | | | | |
| Dimension | 72 mm × 37 mm × 27.3 mm | | | | | | | |
| Type | Solid | | | | | | | |
| Chemical composition | Carbon | Chromium | Iron | Manganese | Molybdenum | Phosphorous | Silicon | Sulfur |
| | 0.380–0.430 | 0.8–1.10 | Balance | 0.75–1.0 | 0.15–0.25 | 0.035 | 0.15–0.30 | 0.040 |
| Hardness | 40 HRC | | | | | | | |
| Tensile strength | 655 MPa | | | | | | | |
| Yield strength | 415 MPa | | | | | | | |
| Hardness, Brinell | 197 | | | | | | | |
| Density | 7.85 g/cm$^3$ | | | | | | | |
| Melting point | 1416 °C | | | | | | | |

Thermocouple and Digital Multimeter: A system composed of a K-type thermocouple and a digital multimeter were employed to determine the contact temperature between the grinding wheel and the workpiece.

### 2.1.1. Experimental Set Up

The grinding machine used in this experiment was an Okamoto (Woodlands Newtown, Singapore) hydraulic surface grinder (Model: PSG-125) with a driving power of main motor 10 HP. It was a column type surface grinding machine and originated in Japan. The optimal position of the nozzle during flood cooling was such that the jet became tangent to the wheel and impinged on the contact region between the workpiece and grinding wheel. To achieve this, the nozzle was placed at a 15° angle with the ground and at a 35 mm distance, which was kept constant. For better accuracy, prior to each experimental run the grinding wheel was dressed with a single-point diamond dresser with a total depth of dressing 40 μm along with a peripheral wheel speed of 15 m/s and feed rate of 400 mm/min, respectively. Each experimental run was conducted twice to check the consistency of the result. The grinding condition, presented in Table 3, incorporated specifications of grinding conditions along with dressing cut parameters.

**Table 3.** Grinding conditions.

| | | |
|---|---|---|
| | Grinding mode | Single pass surface grinding, down cut |
| | Grinding machine | Okamoto hydraulic surface grinder |
| | Nozzle angle | 15° |
| | Flow rate | 5 L/min |
| Grinding Conditions with Specification | Flow measuring device | Z-5615 Panel Flowmeter |
| | Machining condition | Dry, Wet (Flood) |
| | Labor cost | 0.06 BDT/s |
| | Power consumption cost | 8 BDT/unit |
| | Cutting fluid cost | 0.5 BDT/s |
| | Dresser | Single point diamond, HS050 |
| Dressing Cut Parameter | Total depth of dressing | 300 μm |
| | Dressing speed | 400 mm/min |
| | Grinding wheel wear per dress, radially | 3.75 μm |

The input parameters with their respective levels are shown in Table 4. Here, grinding wheel speed was set at 15–25 m/s which is analogous to some of the previous values found in the literature. Khan et al. set the cutting speed at 15–35 m/s to optimize output responses for grinding of AISI D2 steel [43]. Gholami and Azizi set the wheel speed at 1963–2749 m/min (32–45 m/s) for grinding of 1.2080 steel while optimizing time, cost, and surface roughness [28]. Rabiei et al. set the wheel speed at 30 m/s for 100Cr6 hardened steel to improve the performance of the MQL technique [41]. Singh et al. put the wheel speed at 22 m/s to optimize the output responses at different lubricating methods in the surface grinding of 'Ti-6Al-4V-ELI' [38]. Input parameters, such as workpiece velocity, wheel velocity, cross-feed, and cutting condition, have two levels each (Figure 2a), while depth of cut has five levels. The multilevel full factorial design ($2^4 \times 5 = 80$) was applied to run 80 experiments for this research work. Figure 2b shows the multilevel full factorial design of the design of experiments (DOE) applied to conduct experiments. Detailed calculations of the DOE are presented in the Supplementary Materials.

For this surface grinding operation, a temporary setup is arranged to determine the contact temperature between the grinding wheel and the workpiece. This system is composed of a K-type thermocouple and a digital multi-meter connected systematically. The schematic diagram of the experimental setup is depicted in Figure 2a.

**Table 4.** Input parameters and their levels.

| SI. No. | Grinding Input Parameters | Level 1 | Level 2 | Level 3 | Level 4 | Level 5 |
|---------|---------------------------|---------|---------|---------|---------|---------|
| 1 | Workpiece velocity, $v_w$ (m/min) | 5 | 12 | - | - | - |
| 2 | Wheel velocity, $v_s$ (m/s) | 15 | 25 | - | - | - |
| 3 | Depth of cut, $a_p$ (mm) | 0.07 | 0.08 | 0.09 | 0.095 | 0.10 |
| 4 | Cross-feed, $f_b$ (mm/pass) | 3 | 6 | - | - | - |
| 5 | Cutting condition | Dry | Wet | - | - | - |

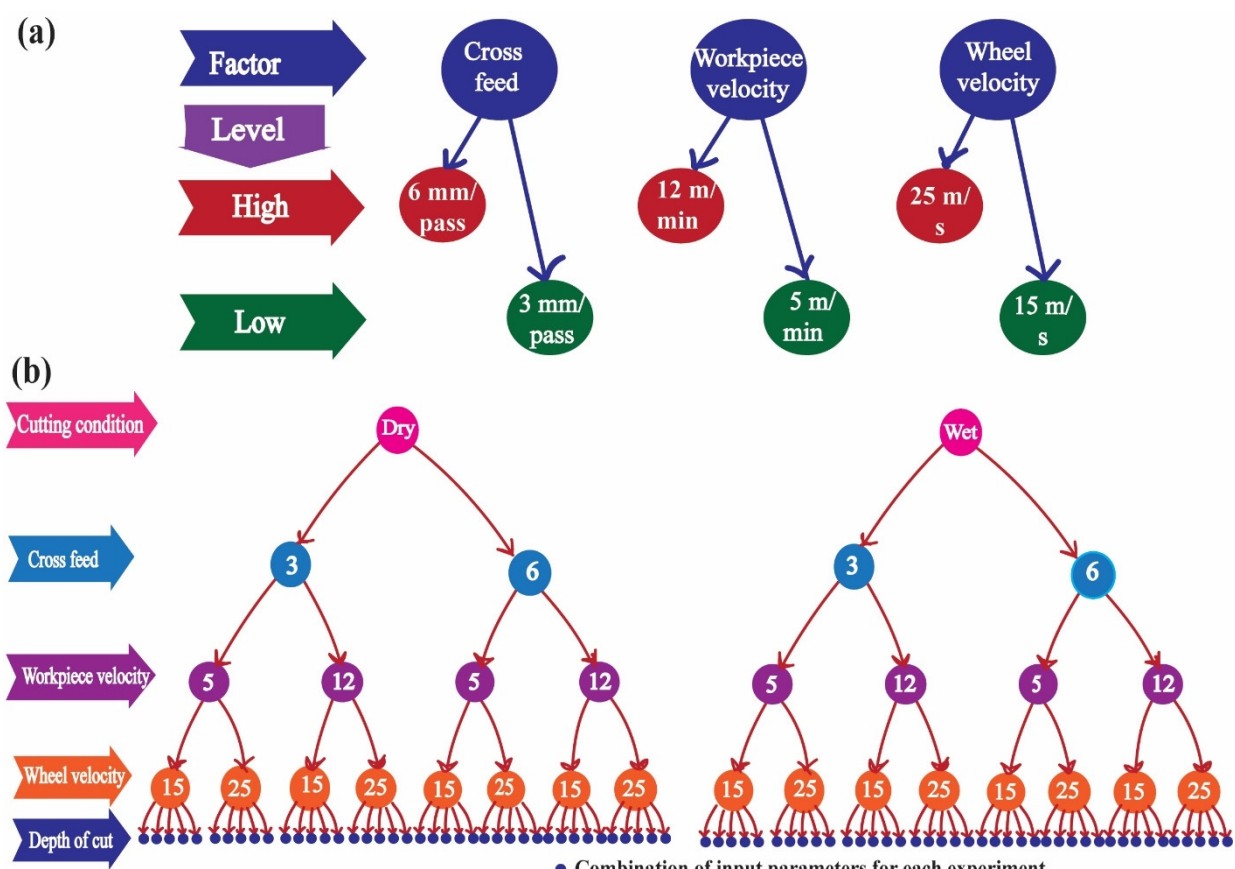

**Figure 2.** A schematic diagram depicting the design of experiment (DOE) conducted for this research work. (**a**) shows high low level of the parameters and (**b**) shows the multilevel full factorial design ($2^4 \times 5 = 80$) applied to run 80 experiments.

Close to the cutting zone, EMF (electromotive force) is generated at the thermocouple junction which is determined from the multi-meter at the other pole of the thermocouple. The embedded thermocouple was inserted into the material to measure the temperature. For this insertion, a very small slot was created by matching thermocouple dimensions. The smaller the size of the slot for positioning thermocouple, the more accurate the data as it minimized the transmission of disturbance to the local temperature field. Hence, a slot of 0.5 mm width and 8 mm depth was made by the EDM (Electrical Discharge Machining) wire cut machine (Figure 3a) on the workpiece and the thermocouples' junction was installed into the slot to make sure that it remains very close to the contact region of grinding wheel and workpiece (Figure 3b). Hence this method enabled a direct temperature measurement of the workpiece surface. Thermocouple junction into the slot was insulated

to minimize convection and radiation heat transfer. The other junction of the thermocouple, which had positive and negative terminals, was connected with the corresponding points of the digital multi-meter. The thermocouple measured the EMF generated due to the presence of the temperature gradient during machining which was transmitted through the thermocouple and displayed at the digital multi-meter. This EMF was converted to the temperature using the following equation to get the elevated temperature at the contact zone of the grinding wheel and the workpiece:

$$\theta(°C) = 75.28 + 63.05v(\text{mV}) - 0.57v^2(\text{mV})$$

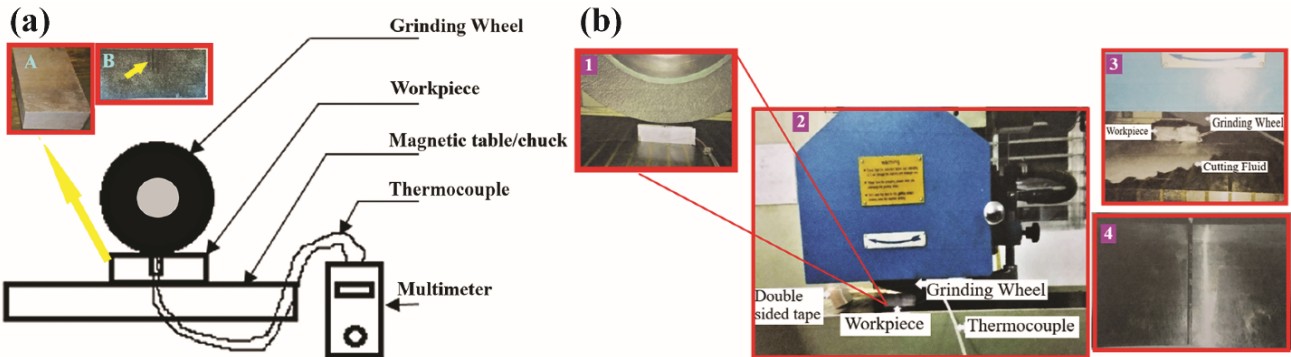

**Figure 3.** (**a**) Schematic diagram of the experimental setup where inset A and B shows photograph images workpiece before slot cutting and after slot cutting using wire EDM, and (**b**) pictorial view of the experimental setup. A—Workpiece before machining. B—Wire cut on the workpiece. 1—Thermocouple setup on the workpiece. 2—Dry machining setup. 3—Wet machining setup. 4—Workpiece after machining.

Cutting zone temperature in terms of digital multi-meter EMF reading was obtained by varying the cutting parameters, including spindle speed, feed, and depth of cut for different cutting conditions (dry condition (Figure 3b) and wet condition (Figure 3b)). To calculate *MRR*, material removed after each pass of machining per minute for a particular set of input parameters was determined. This action was carried out through slide calipers which measured the reduction of height of the workpiece after each round of machining. This reduction of height was multiplied with the length and width of the workpiece to obtain the total amount of material removed. A stopwatch was used to determine the time for each round of machining. Total amount of material removed was divided by total machining time for each round to get experimental *MRR*. Total cost included utility (power consumption), labor, and cutting fluid cost for this research work. To calculate utility cost, power consumption for each round of machining was determined, which was further multiplied with cost per unit of consumption. Labor and cutting fluid costs were obtained by multiplying the machining time for each round with cost per unit time. For each set of input parameters, output responses were measured thrice, and their geometric means were used for further mathematical analysis. The measurement procedure was a continuous procedure and measured at a single set up. For each set of input parameters, a particular round of machining was performed (each rounds repeated thrice) and these rounds were performed sequentially for all sets of input parameters. Each round was followed by the measurement of responses. Response surface methodology (RSM) and desirability function approach (DFA) were used as the statistical methods to process the experimental results. Under this arrangement, surface grinding operation was conducted in a horizontal surface grinding machine. The machined surface of the workpiece looks like that shown in Figure 3b after the operation.

To optimize the surface grinding process, response surface methodology (RSM) and desirability function approach (DFA) were employed in the solution methodology. Response

surface methodology and desirability function approach are detailed in the subsequent sub-sections.

### 2.1.2. Material Removal Rate (*MRR*) Calculation

The amount of material removed per unit time (theoretically) from the workpiece by machining is determined by the following equation [20]:

$$MRR = 100 \times v_w \times a_p \times b \tag{1}$$

where:

$MRR$ is the material removal rate (mm$^3$/min)
$v_w$ is the workpiece velocity or longitudinal table travel velocity (m/min)
$a_p$ is the depth of cut (mm)
$b$ is the width of the cut (mm).

### 2.1.3. Response Surface Methodology

Design of experiments is most commonly adopted to investigate the impact of input variables on the output responses at various fields including machining parameter optimization. An optimal set of input variables can be found systematically with fewer trials, effort, and time by this process. Response surface methodology (RSM) is one of the most popular methods among the design of experimental methods. RSM is an empirical model developing method with mathematical and statistical analysis (especially regression analysis) to optimize an output response (dependent variable) which is regulated by several input parameters (independent variables). A comprehensive outline of RSM methodology is depicted in Figure 4. Coefficients of the generalized quadratic model (b$_0$, b$_i$, b$_{ii}$, and b$_{ij}$) were calculated, based on the least-squares method [44].

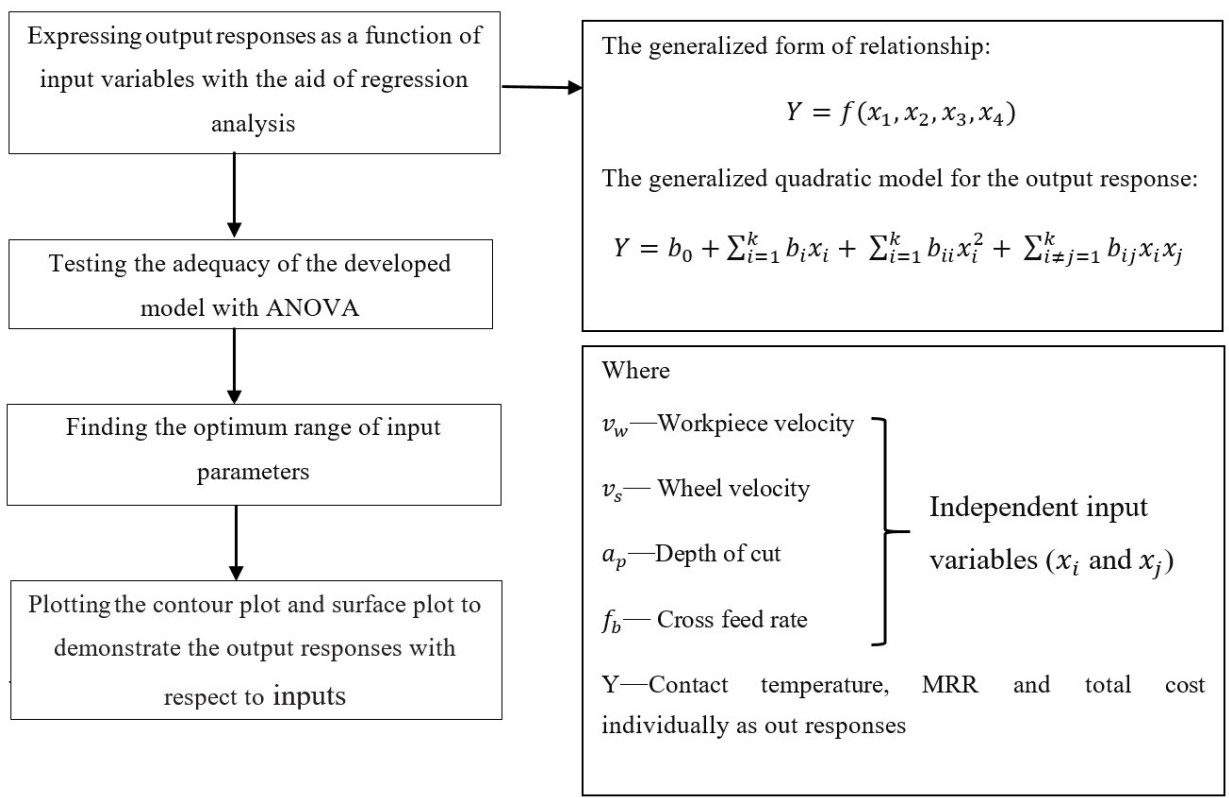

**Figure 4.** A flowchart showing RSM methodology.

RSM models mathematically how two or more independent parameters regulate the output response of a process or system and also find the optimal set of independent parameters. With the aid of ANOVA significant parameters (which influence the output response) are screened out from the non-significant parameters [45–47].

2.1.4. Desirability Function Approach

RSM focuses on the optimization where there is only one output response. However, in the actual scenario of process optimization, there is more than one output response of interest, and these responses may be contradictory. In this case, to determine the optimum set of input parameters, all the output responses should be taken under consideration at the same time. This scenario is termed as a multi-response problem or multi-response optimization (MRO) problem [48]. In this study optimization of contact temperature, *MRR*, and machining cost are the key objectives, but these output responses have a conflicting interest as contact temperature and machining cost need to be minimized while *MRR* needs to be maximized. Since multiple conflicting objectives are present, this is termed as a multi-response optimization (MRO) problem. So far, to solve MRO problems numerous methods have been developed, i.e., desirability function approach [49,50] and loss function approach [51].

Among the MRO techniques, DFA is very simple to implement and easy-to-understand approach for the industry practitioners. The underlying principle of DFA is to convert a multi-objective problem into a single objective problem with the aid of mathematical conversions. Derringer and Suich (1980) demonstrated a multiple objective problem with the help of desirability-function to make it popular for industry practitioners for optimizing the multiple quality characteristic problems [52]. DFA deals with an objective function, named desirability function, and converts the calculated output from the desirability function into a scale-free value called desirability with one out of three cases. In this work a multi response optimization with DFA was used and the equations showed in the DFA flowchart (Figure 5) are reported by Manohar et al. and the steps are illustrated as flowchart in Figure 5 [20].

$$X_{ik} = (x_{i1}, x_{i2}, x_{i3}, x_{i4}) = (v_{si}, v_{wi}, d_i, f_{bi}) \tag{2}$$

where:

$X_i$—input variables value for *i* in the experiment, $I = 1, 2, \ldots, m$.

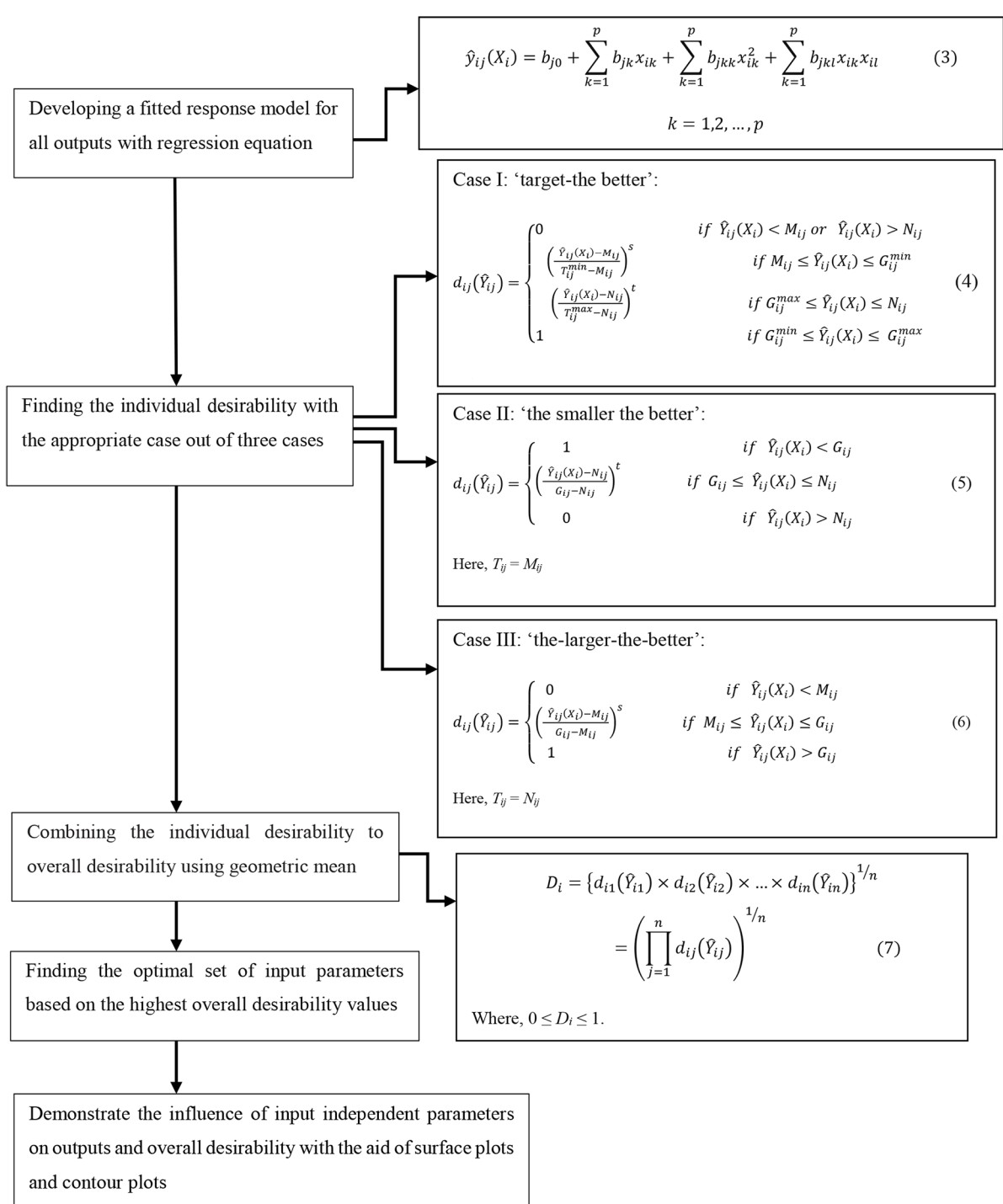

**Figure 5.** Flowchart of DFA methodology.

## 3. Results and Discussions

The main focuses of this study were solving the multi-response optimization problem in surface grinding of AISI 4140 alloy steel with the aim of maximizing the desirability function. Three responses, temperature, material removal rate (*MRR*), and total machining cost, were to be optimized with respect to cross-feed, workpiece velocity, the wheel or cutting velocity, and the depth of cut under dry and wet conditions. It can be found that the wheel velocity and the workpiece velocity of this study are analogous to the input parameters of the study reported by Sadeghi et al. [53]. However, they used a depth of cut which was lower than that in our study due to the use of AISI 4140 hardened steel instead

of regular AISI 4140 steel (used in our study). Sadeghi et al. studied the advantages of MQL technique through the evaluation of grinding forces and surface quality for grinding AISI 4140 hardened steel. They set the input parameters wheel speed at 30 m/s, work speed at 10, 20, 30, and 40 m/min, depth of cut at 0.005, 0.010, and 0.015 mm [53]. Response surface methodology (RSM) was applied to model the responses for prediction whereas the desirability function approach (DFA) was employed to resolve the multi-objective optimization problem. The use of DFA also enabled the assignment of numerous weights to the responses according to their relative importance. Analysis of variance (ANOVA) was conducted to confirm the model adequacy. For equal weights of responses, the optimal values were found to be 6 mm/pass of cross-feed, 12 m/min of workpiece velocity, 15 m/s of the wheel or cutting velocity, 0.095 mm of the depth of cut in wet condition, and a maximum overall desirability value of 0.863. Additional results were obtained for the unequal weights of response variables. Surface and contour plots were also developed for further examination of the obtained result. The estimated equation for total cost and temperature by RSM under different environments (dry and wet) is given below:

$$
\begin{aligned}
\text{Temp}_{\text{dry}} = \ & 977.242 + 7.25 \times f_b - 3.36 \times v_w + 2.375 \times v_s - 21602.5 \times a_p \\
& + 145538 \times a_p^2 - 0.212 \times f_b \times v_w - 0.071 \times f_b \times v_s \\
& - 28.223 \times f_b \times a_p - 0.0042 \times v_w \times v_s \\
& + 23.7174 \times v_w \times a_p - 8.89 \times v_s \times a_p
\end{aligned}
\tag{8}
$$

$$
\begin{aligned}
\text{Cost}_{\text{dry}} = \ & 9.4224 - 0.978 \times f_b - 0.49943 \times v_w + 0.0043143 \times v_s \\
& - 22.83 \times a_p + 97.1215 \times a_p^2 + 0.0534 \times f_b \times v_w \\
& - 0.00007 \times f_b \times v_s + 1.434 \times f_b \times a_p \\
& + 0.000157143 \times v_w \times v_s + 0.1429 \times v_w \times a_p \\
& - 0.01552 \times v_s \times a_p
\end{aligned}
\tag{9}
$$

$$
\begin{aligned}
\text{Temp}_{\text{wet}} = \ & 423.585 + 4.63\text{E} - 14 \times f_b - 0.4011 \times v_w - 1.44345 \times v_s \\
& - 7958.2 \times a_p + 52175.4 \times a_p^2 - 2.78\text{E} - 16 \times f_b \times v_w \\
& - 3.11\text{E} - 16 \times f_b \times v_s - 4.36\text{E} - 13 \times f_b \times a_p \\
& - 0.0137251 \times v_w \times v_s - 7.89962 \times v_w \times a_p \\
& + 23.8998 \times v_s \times a_p
\end{aligned}
\tag{10}
$$

$$
\begin{aligned}
\text{Cost}_{\text{wet}} = \ & 39.7315 - 5.84 \times f_b - 3.1523 \times v_w + 0.2445 \times v_s + 344.873 \times a_p \\
& - 1587.53 \times a_p^2 + 0.354143 \times f_b \times v_w + 0.01057 \times f_b \times v_s \\
& - 5.1552 \times f_b \times a_p - 0.01133 \times v_w \times v_s \\
& + 0.1195 \times v_w \times a_p - 2.25172 \times v_s \times a_p
\end{aligned}
\tag{11}
$$

From the equations, it is obvious that the temperature and the total cost were affected by the depth of cut more significantly than any other input parameters, as the coefficient of the depth of cut was much higher than any other coefficients in the equation. In the RSM, each estimated equation was developed considering values of 40 sets of input variables. Afterward, these equations were used to predict the responses. It was found that the predicted and experimental values are quite proximate. The mean error for temperature and the total cost was less than 2% and 5%, respectively. This depicts that the experiments were conducted in the right way and uncertainty in input parameters was lower. After that, response surfaces for each combination of input parameters were constructed, as shown in Figures 5 and 6. For the dry condition (Figure 6a), the optimum levels for workpiece velocity, wheel velocity, depth of cut, and cross-feed were 12–20 m/min, 2–10 m/s, 0.065–0.085 mm, and 0.1–2.5 mm/pass, respectively, to minimize the temperature. For the wet condition (Figure 6b), the optimum levels for workpiece velocity, wheel velocity, and depth of cut were 15–20 m/min, 1–12 m/s, and 0.062–0.080 mm, respectively, to minimize the temperature. However, cross-feed had no significant effect on temperature in wet the condition. Similarly, the response of total cost was analyzed in this way and optimum levels were determined from contour plots (Figure 7). The results are represented in Table 5.

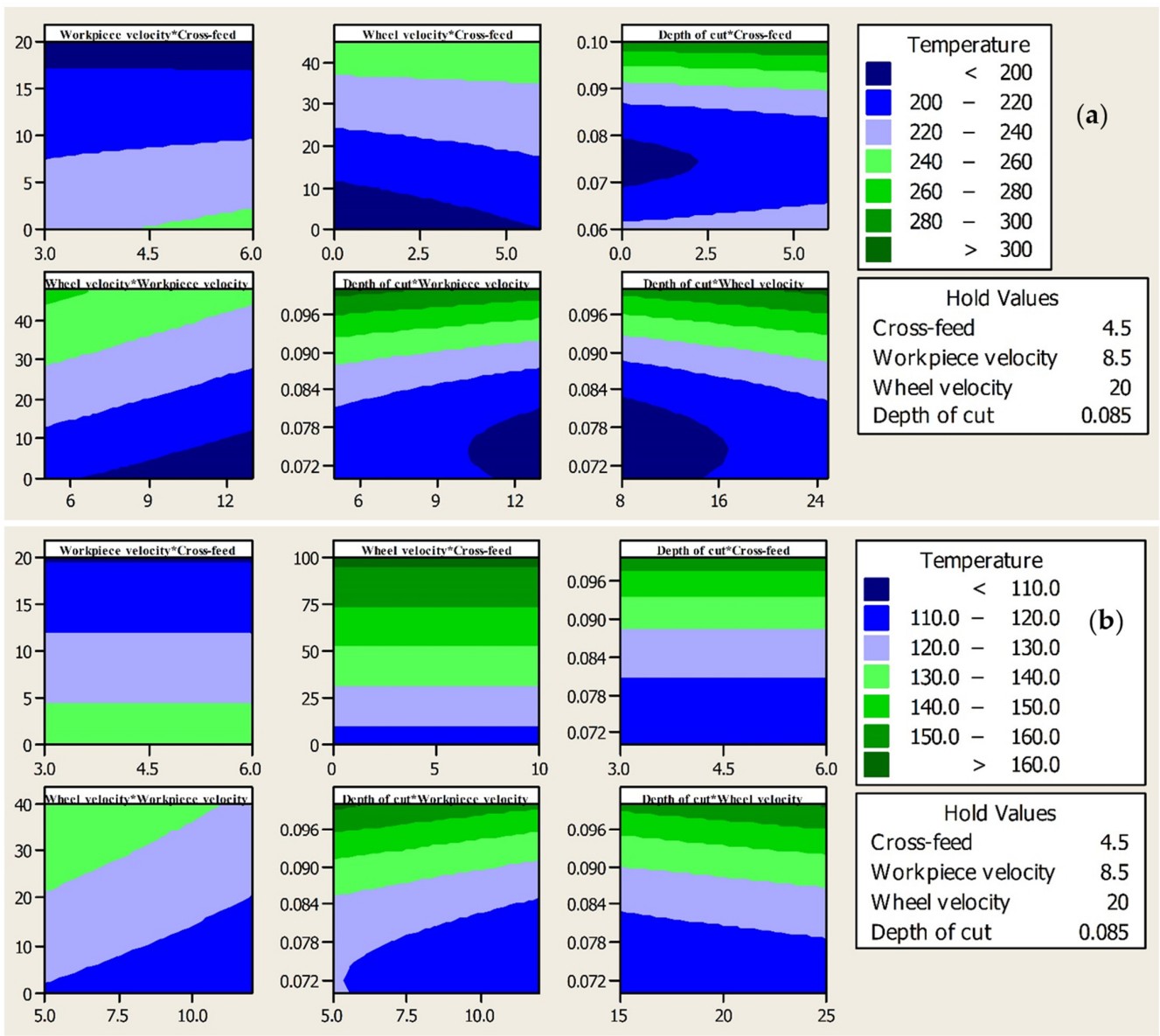

**Figure 6.** Contour plot of the temperature for each combination of input parameters. (**a**) Surface grinding under dry condition and (**b**) surface grinding under wet condition.

**Table 5.** Optimum range of input parameters from RSM.

| Objectives | Parameters/ Condition | Workpiece Velocity, $v_w$ (m/min) | Wheel Velocity, $v_s$ (m/s) | Depth of Cut, $a_p$ mm | Cross-Feed, $f_b$ (mm/Pass) |
|---|---|---|---|---|---|
| The optimum level for temp | Dry | 12–20 | 2–10 | 0.065–0.085 | 0.1–2.5 |
| The optimum level for the cost | | 12–15 | - | 0.03–0.13 | 6.5–8 |
| The optimum level for temp | Wet | 15–20 | 1–12 | 0.062–0.080 | - |
| The optimum level for the cost | | 15–20 | - | - | 7–10 |

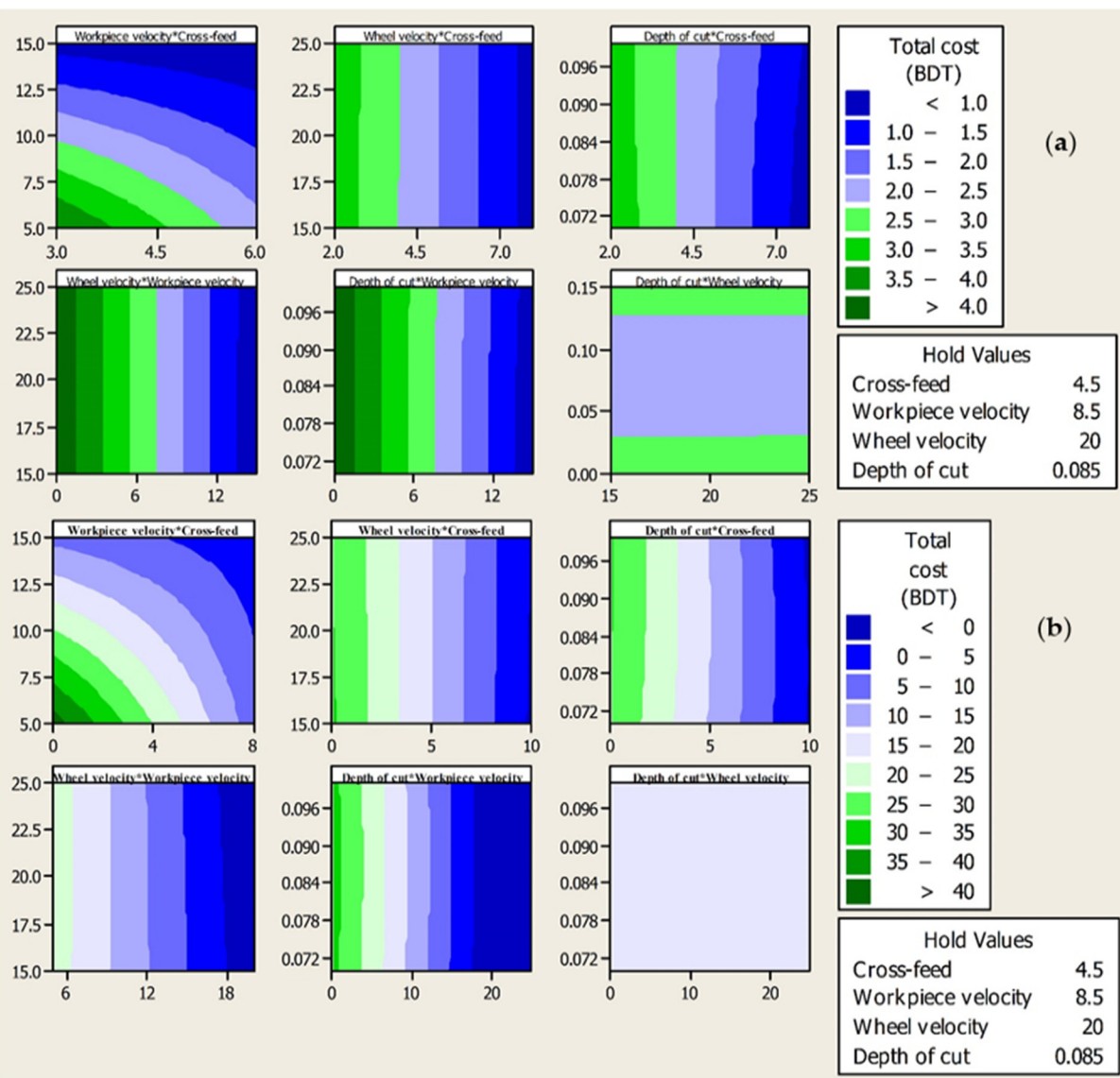

**Figure 7.** Contour plot of the total cost for each combination of input parameters. (**a**) Surface grinding under dry condition and (**b**) surface grinding under wet condition.

To carry out the experiments, target values for all independent variables (input parameters) and three dependent variables (response) were selected. The optimization criteria for these two sets of variables are presented in Table 6. Target values for the independent variables should lie within the range of the parameter settings that were used to conduct the experiments. The input parameter's range for optimization was determined based on the optimum range of input parameters derived from RSM contour plot and from previous literature. Luu Anh Tung et al. set table speed (m/min) at four levels [6,8,10,12] to investigate its influence on surface roughness of grinding 9CrSi tool steel [50]. Kruszynski and Lajmert set cross-feed (mm) at three levels [2,4,8] to maximize *MRR* during grinding of 34CrAl6C steel [51]. Again, depth of cut (mm) and wheel speed (m/min) were set at three levels (0.06, 0.08, 0.1) and (1963, 2356, 2749), respectively, for time, cost, and surface roughness optimization during 1.2080 steel grinding by Mohammad Hadi Gholami and Mahmood Reza Azizi [52]. Besides, for optimum production cost, production rate and surface finish the wheel speed (m/min) were 2000 and 1998, respectively, by quadratic programming and genetic algorithm for rough grinding, and 2000 and 1986, respectively, by quadratic programming and genetic algorithm for finish grinding at a surface grinding

operation by Saravanan et al. [53]. The wheel velocity from literature and that we have considered are quite analogous, as it is in m/min units in the literature, but we used the unit m/s. Since the cutting condition is a categorical variable (i.e., either dry or wet), it does not have any lower or upper bound. Among the dependent variables, the target for temperature should be to minimize it since the higher temperature has an increasingly adverse effect on the surface finish and other material characteristics of the workpiece. Total cost, which includes utility cost, labor cost, and cutting fluid cost, should also be minimized. The *MRR* should be maximized because higher *MRR* results in the lower unit production cost of the produce. The lower and higher levels of these variables were set to the minimum and maximum values that were observed in the experiment. Alternatively, one can use more relaxed or stringent bounds. Regardless, the overall outcome of DFA should remain the same.

**Table 6.** Optimization criteria for input parameters.

| Variable Type | Variables | Target | Lower Bound | Upper Bound |
|---|---|---|---|---|
| Independent (input parameters) | Cutting condition | Dry or Wet | N/A | N/A |
| | Cross-feed (mm/pass) | In range | 3 | 6 |
| | Workpiece velocity (m/min) | In range | 5 | 12 |
| | Wheel velocity (m/s) | In range | 15 | 25 |
| | Depth of cut (mm) | In range | 0.07 | 0.10 |
| Dependent (response) | Temperature (°C) | Minimize | 109.78 | 313.67 |
| | *MRR* (mm$^3$/min) | Maximize | 1050 | 7200 |
| | Total cost (BDT) | Minimize | 0.91 | 29.48 |

The individual desirability values were calculated according to Equations (4)–(6) with DFA as depicted at Figure 5. The individual desirability values were calculated for each experimental run. The overall desirability value was then the geometric mean of these individual desirability values when the weights were all equal. The individual desirability values and overall desirability values for three schemes are listed at Table 7. However, we also considered some cases when different weights were applied to the quality factors. Using Equation (7), we considered two such cases and calculated the values as shown in Table 7. The weights are shown in terms of percentage, i.e., 15:25:60 represents that 15%, 25%, and 60% weightage are assigned to temperature, *MRR*, and total machining cost, respectively. This scheme can be highly advantageous in situations when one particular response might need higher attention than the other responses. For instance, it might be in the interest of the production engineer to achieve high *MRR* to speed up the grinding or any other machining process. This sudden change of interest can be easily accommodated in the desirability function by assigning higher weights to *MRR*. Therefore, this method is highly flexible in the sense that there is always an option at hand to focus on a particular response if needed. However, the most desirable value for each scheme is highlighted at Table 8. The experimental run at this desirability value indicates the optimum set of input parameters and output responses.

**Table 7.** Individual and overall desirability values for three different cases.

| Individual Desirability | | | Overall Desirability (Equal Weightage 33:33:33) | Overall Desirability (20:40:40) | Overall Desirability (15:25:60) |
|---|---|---|---|---|---|
| Temp. | *MRR* | Total Cost | | | |
| 0.533 | 0 | 0.890 | 0 | 0 | 0 |
| 0.527 | 0.025 | 0.891 | 0.228 | 0.193 | 0.338 |
| 0.377 | 0.049 | 0.891 | 0.255 | 0.236 | 0.380 |
| 0.249 | 0.061 | 0.891 | 0.239 | 0.237 | 0.377 |
| 0.086 | 0.074 | 0.891 | 0.179 | 0.207 | 0.337 |
| 0.458 | 0 | 0.888 | 0 | 0 | 0 |
| 0.456 | 0.025 | 0.890 | 0.217 | 0.187 | 0.330 |
| 0.312 | 0.049 | 0.890 | 0.240 | 0.227 | 0.369 |
| 0.186 | 0.061 | 0.890 | 0.217 | 0.223 | 0.361 |
| 0.024 | 0.074 | 0.890 | 0.117 | 0.160 | 0.278 |
| 0.615 | 0.240 | 0.969 | 0.524 | 0.507 | 0.639 |
| 0.601 | 0.298 | 0.970 | 0.559 | 0.550 | 0.673 |
| 0.444 | 0.357 | 0.970 | 0.537 | 0.557 | 0.672 |
| 0.311 | 0.386 | 0.970 | 0.489 | 0.535 | 0.650 |
| 0.144 | 0.415 | 0.970 | 0.388 | 0.472 | 0.590 |
| 0.542 | 0.240 | 0.968 | 0.502 | 0.494 | 0.627 |
| 0.532 | 0.298 | 0.969 | 0.536 | 0.537 | 0.660 |
| 0.379 | 0.357 | 0.969 | 0.509 | 0.539 | 0.656 |
| 0.249 | 0.386 | 0.969 | 0.454 | 0.511 | 0.628 |
| 0.084 | 0.415 | 0.969 | 0.324 | 0.424 | 0.544 |
| 0.487 | 0.171 | 0.954 | 0.431 | 0.420 | 0.562 |
| 0.484 | 0.220 | 0.953 | 0.467 | 0.463 | 0.597 |
| 0.339 | 0.269 | 0.953 | 0.444 | 0.468 | 0.595 |
| 0.213 | 0.293 | 0.951 | 0.391 | 0.441 | 0.567 |
| 0.052 | 0.318 | 0.951 | 0.251 | 0.344 | 0.468 |
| 0.423 | 0.171 | 0.953 | 0.411 | 0.408 | 0.550 |
| 0.425 | 0.220 | 0.952 | 0.447 | 0.451 | 0.585 |
| 0.284 | 0.269 | 0.951 | 0.418 | 0.451 | 0.579 |
| 0.160 | 0.293 | 0.950 | 0.355 | 0.416 | 0.542 |
| 0.001 | 0.318 | 0.950 | 0.068 | 0.156 | 0.259 |
| 0.591 | 0.649 | 0.995 | 0.726 | 0.756 | 0.827 |
| 0.580 | 0.766 | 0.994 | 0.762 | 0.805 | 0.860 |
| 0.427 | 0.883 | 0.992 | 0.721 | 0.800 | 0.850 |
| 0.297 | 0.942 | 0.992 | 0.653 | 0.764 | 0.818 |
| 0.132 | 1.000 | 0.991 | 0.508 | 0.665 | 0.735 |
| 0.528 | 0.649 | 0.993 | 0.699 | 0.739 | 0.813 |
| 0.522 | 0.766 | 0.992 | 0.735 | 0.787 | 0.845 |
| 0.373 | 0.883 | 0.991 | 0.689 | 0.779 | 0.832 |
| 0.246 | 0.942 | 0.990 | 0.613 | 0.735 | 0.794 |

**Table 7.** *Cont.*

| Individual Desirability | | | Overall Desirability (Equal Weightage 33:33:33) | Overall Desirability (20:40:40) | Overall Desirability (15:25:60) |
|---|---|---|---|---|---|
| Temp. | MRR | Total Cost | | | |
| 0.082 | 1.000 | 0.989 | 0.434 | 0.604 | 0.683 |
| 0.951 | 0 | 0.052 | 0 | 0 | 0 |
| 0.942 | 0.025 | 0.031 | 0.091 | 0.057 | 0.050 |
| 0.882 | 0.049 | 0.022 | 0.099 | 0.064 | 0.047 |
| 0.832 | 0.061 | 0.022 | 0.104 | 0.069 | 0.049 |
| 0.770 | 0.074 | 0.024 | 0.112 | 0.076 | 0.054 |
| 0.943 | 0 | 0.030 | 0 | 0 | 0 |
| 0.923 | 0.025 | 0.018 | 0.075 | 0.046 | 0.036 |
| 0.850 | 0.049 | 0.016 | 0.088 | 0.056 | 0.039 |
| 0.795 | 0.061 | 0.020 | 0.100 | 0.066 | 0.046 |
| 0.727 | 0.074 | 0.026 | 0.113 | 0.077 | 0.056 |
| 0.991 | 0.240 | 0.603 | 0.524 | 0.461 | 0.517 |
| 0.985 | 0.298 | 0.583 | 0.556 | 0.496 | 0.534 |
| 0.927 | 0.357 | 0.573 | 0.575 | 0.523 | 0.548 |
| 0.879 | 0.386 | 0.572 | 0.580 | 0.533 | 0.553 |
| 0.818 | 0.415 | 0.575 | 0.581 | 0.542 | 0.559 |
| 0.988 | 0.240 | 0.610 | 0.526 | 0.463 | 0.520 |
| 0.970 | 0.298 | 0.597 | 0.558 | 0.499 | 0.540 |
| 0.900 | 0.357 | 0.595 | 0.577 | 0.527 | 0.558 |
| 0.847 | 0.386 | 0.598 | 0.581 | 0.539 | 0.565 |
| 0.780 | 0.415 | 0.605 | 0.582 | 0.548 | 0.572 |
| 0.951 | 0.171 | 0.500 | 0.434 | 0.371 | 0.422 |
| 0.942 | 0.220 | 0.485 | 0.466 | 0.404 | 0.440 |
| 0.882 | 0.269 | 0.481 | 0.486 | 0.431 | 0.456 |
| 0.832 | 0.293 | 0.484 | 0.491 | 0.442 | 0.464 |
| 0.770 | 0.318 | 0.489 | 0.494 | 0.451 | 0.471 |
| 0.943 | 0.171 | 0.467 | 0.423 | 0.360 | 0.404 |
| 0.923 | 0.220 | 0.460 | 0.455 | 0.394 | 0.425 |
| 0.850 | 0.269 | 0.465 | 0.475 | 0.422 | 0.444 |
| 0.795 | 0.293 | 0.471 | 0.480 | 0.433 | 0.453 |
| 0.727 | 0.318 | 0.480 | 0.481 | 0.443 | 0.461 |
| 0.991 | 0.649 | 0.792 | 0.799 | 0.765 | 0.780 |
| 0.985 | 0.766 | 0.776 | 0.837 | 0.810 | 0.802 |
| 0.927 | 0.883 | 0.772 | 0.859 | 0.845 | 0.821 |
| 0.879 | 0.942 | 0.774 | 0.863 | 0.859 | 0.829 |
| 0.818 | 1.000 | 0.779 | 0.861 | 0.870 | 0.836 |
| 0.988 | 0.649 | 0.787 | 0.797 | 0.763 | 0.776 |
| 0.970 | 0.766 | 0.779 | 0.834 | 0.809 | 0.802 |

**Table 7.** *Cont.*

| | Individual Desirability | | Overall Desirability (Equal Weightage 33:33:33) | Overall Desirability (20:40:40) | Overall Desirability (15:25:60) |
|---|---|---|---|---|---|
| **Temp.** | *MRR* | **Total Cost** | | | |
| 0.900 | 0.883 | 0.783 | 0.854 | 0.845 | 0.824 |
| 0.847 | 0.942 | 0.789 | 0.858 | 0.860 | 0.834 |
| 0.780 | 1.000 | 0.798 | 0.854 | 0.870 | 0.842 |

**Table 8.** Optimum cutting parameters and corresponding responses for different weights.

| Weights (%) | Cutting Condition | Cross-Feed (mm/pass) | Workpiece Velocity (m/min) | Wheel Velocity (m/s) | Depth of Cut (mm) | Temp. (°C) | *MRR* (mm³/min) | Total Cost (BDT) | Overall Desirability |
|---|---|---|---|---|---|---|---|---|---|
| 33.33:33.33:33.33 (Equal) | Wet | 6 | 12 | 15 | 0.095 | 140.854 | 6840 | 8.36 | 0.863 |
| 20:40:40 | Wet | 6 | 12 | 15 | 0.100 | 143.945 | 7200 | 7.77 | 0.870 (tied) |
| | | 6 | 12 | 25 | 0.100 | 150.119 | 7200 | 6.32 | |
| 15:25:60 | Dry | 6 | 12 | 15 | 0.080 | 204.565 | 5760 | 1.01 | 0.860 |
| 100:0:0 (temperature only) | Wet | 3 | 12 | 15 | 0.070 | 109.785 | 2520 | 12.24 | 0.991 |
| 0:0:100 (total cost only) | Dry | 6 | 12 | 15 | 0.070 | 191.799 | 5040 | 1.08 | 0.995 |

For different weights, the optimum sets of the cutting parameters and their corresponding temperature, *MRR*, and total cost values are summarized in Table 8. The first row represents the scenario when all quality factors are given equal importance. It is evident that if the temperature is prioritized, then the wet condition dominates the overall desirability, which is expected. However, if the cost is given high weightage, dry condition achieves the highest overall desirability. This is primarily because cutting in the wet condition is more expensive since cutting fluid has to be used. In general, the high value of cross-feed and workpiece velocity, while a low-level value of wheel velocity dominates the optimum result in the cases shown in Table 7. However, if different combinations of weights are assigned, this scenario will most possibly change.

We also included two scenarios when the sole focus is given to only one quality factor, that is, 100% weight is assigned to a particular response. Therefore, if somehow the process diverts from a multi-objective problem to a single-objective problem, we do not need to devise a new method. DFA will still be able to suggest a set of process parameters for the optimum value of the single factor that is under sole consideration. In such cases, DFA will simply apply 100% weight to the factor under consideration and ignore the other factors by assigning 0% weight, as illustrated in Table 8.

Figure 8a shows an individual desirability graph for temperature considering the variation of cross-feed and depth of cut only. It is evident from this figure that increasing the depth of cut has an adverse influence on the individual desirability for temperature. Cross-feed has no effect in this case. Increasing values of both depth of cut and cross-feed result in increasing value of individual desirability for *MRR* as observed from the surface plot in Figure 8b. Since desirability for temperature has a negative relation with the depth of cut while desirability for *MRR* has a positive relation, a conflicting situation is created. Hence, the need for the construction of the overall desirability function is visible. We now plot the overall desirability function instead of individual ones. We consider the scenario where all the responses are given equal importance. Similar insights can be made in the same way for cases where the weights are not equal. For this illustration, one of the parameters (depth of cut in this case) is always kept constant on one axis while other parameters (cross-feed, workpiece velocity, and wheel velocity or cutting velocity) are varied. Being a categorical variable, cutting conditions cannot be used as a parameter.

Rather, a separate analysis should be done to see the effect of varying the other parameters in a particular cutting condition.

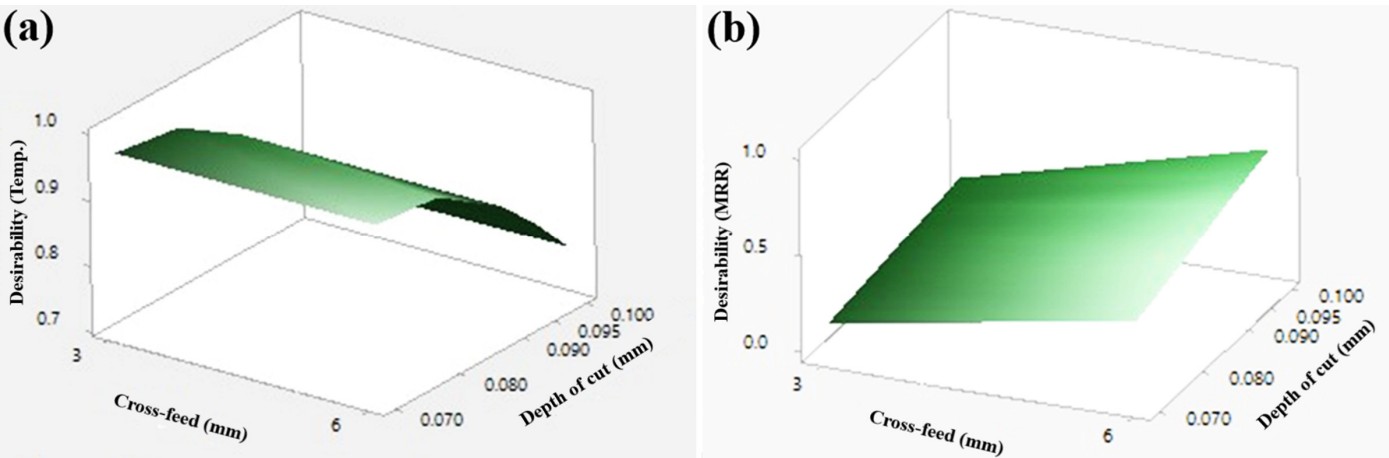

**Figure 8.** Surface plot of individual desirability (**a**) for temperature in wet condition, and (**b**) for *MRR* in wet condition.

Surface plots and contour plots for overall desirability in the dry cutting condition are shown in Figure 9. Keeping the other parameters constant, desirability values increase up to a point (<0.095 mm) with a higher depth of cut, and then decrease in the region where the depth of cut is greater than 0.095 mm. This is consistent across all the surface plots (Figure 9). Again, if the depth of cut is kept constant, cross-feed and workpiece velocity or table velocity show a positive relationship with overall desirability values (Figure 9a,b), while the cutting velocity has no effect whatsoever (Figure 9c). From Figure 9, it is obvious that these observations are consistent with the corresponding contour plots as well.

Surface and contour plots for the wet conditions, shown in Figure 10, reveal that workpiece velocity and cross-feed are positively related to higher overall desirability values (Figure 10a,b). From the surface plots, one might conjecture that the depth of cut also has a positive relationship with overall desirability. However, contour plots indicate that this positive correlation is rather negligible. From surface and contour plots shown in Figure 10c, it is also observed that the effect of cutting velocity, i.e., wheel velocity, on overall desirability is negligible.

The surface and contour plots for a specific cutting condition and weight provide insights about the sensitivity of the overall desirability values for the individual parameters. For example, from the discussion above, it is evident that the depth of cut positively affects the overall desirability values in the dry cutting condition, but has a negligible effect in the wet condition. It must be kept in mind that these conjectures are not universal; rather they depend on how the weights are assigned. Cutting velocity has no apparent effect on desirability when all weights are equal in our case. However, this scenario will most likely change if different weights are assigned.

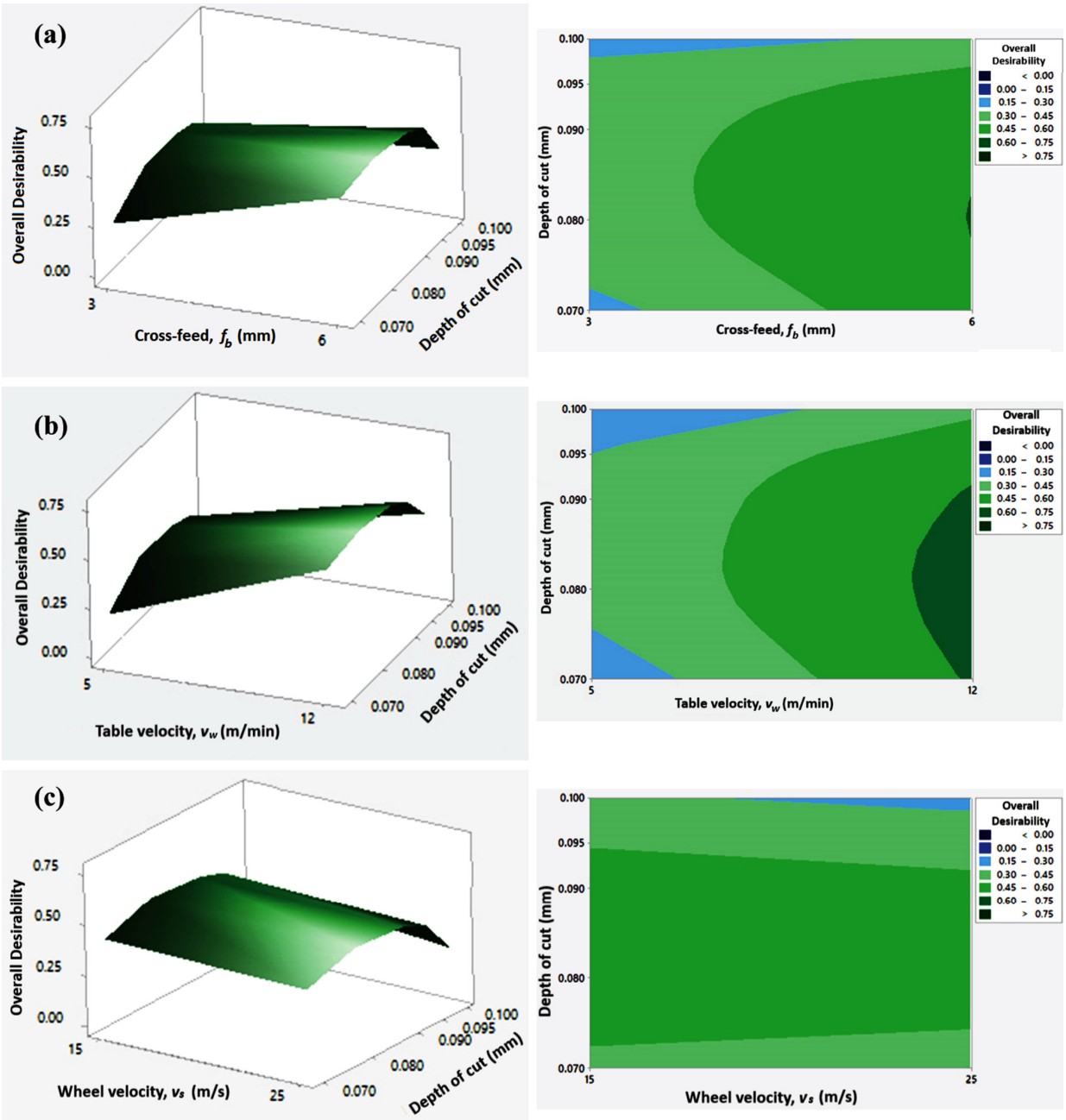

**Figure 9.** Surface plot and corresponding contour plot for overall desirability in dry condition. (**a**) show the combined effect of cross-feed and depth of cut, (**b**) table velocity and depth of cut and (**c**) wheel velocity and depth of cut on overall desirability.

Confirmation tests of optimized parameters i.e., checking the validity of optimal set of predicted responses with the real experimental run has been conducted. Here, improvement of the responses is compared with initial process parameters. The optimal set of input parameters for equal weightage scheme is found at wet cutting condition by 6 mm cross-feed, 12 m/min table velocity, 15 m/s wheel velocity, and 0.095 depth of cut (Table 8). Its corresponding predicted output responses are: 134.555 °C, 7.366 BDT, and 0.844 for temperature, total cost, and overall desirability respectively. On the other hand, experimental value for temperature and total costs are 140.854 °C and 8.36 BDT, respectively, with the overall desirability to be 0.863. Similarly, for rest of the schemes optimal set of input process parameters (Table 9) and their corresponding predicted responses along with

the experimental results are presented at Table 9. Error of the predicted value from the experimental value for equal weightage scheme are as: Temperature is 4.47% and total cost is 7.37% (Table 9). Error for temperature and total cost at rest of the experimental runs are less than 5% and 8% respectively. The resulting deviation between experimental and predicted overall desirability value is negligible (less than 1% except the first case). Therefore, the predicted response and experimental values are quite analogous, and the mathematical model is fair enough to predict the temperature and total cost. Note that we have determined *MRR* analytically using Equation (1). So, there is no error for this parameter.

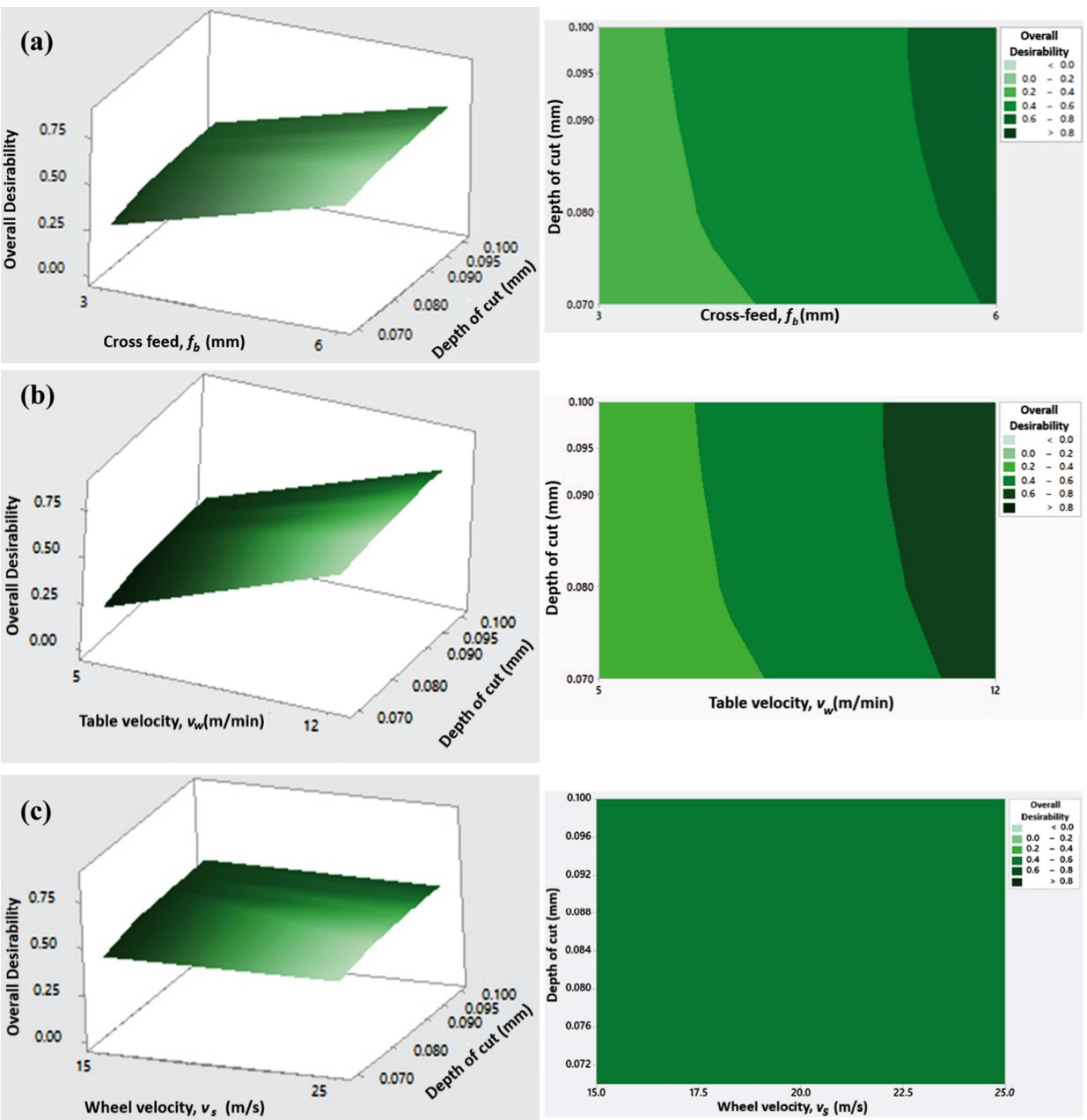

**Figure 10.** Surface plot and corresponding contour plot for overall desirability in wet condition (equal weightage). (**a**) show the combined effect of cross-feed and depth of cut, (**b**) table velocity and depth of cut and (**c**) wheel velocity and depth of cut on overall desirability.

**Table 9.** Comparison of predicted and experimental responses using DFA method.

| Schemes | Cutting Condition | Optimum Cutting Parameters | Optimum Process Parameters | | | | | | Absolute Error (%) | | |
| | | | Experimental Values | | | Predicted Values | | | | | |
| Weights (%) | | | Temp. (°C) | Total Cost (BDT) | Overall Desirability | Temp. (°C) | Total Cost (BDT) | Overall Desirability | Temp. (°C) | Total Cost (BDT) | Overall Desirability |
| 33.33:33.33:33.33 (Equal) | Wet | $(f_b, v_w, v_s, a_p)$ = (6, 12, 15, 0.095) | 140.854 | 8.36 | 0.863 | 134.555 | 7.366 | 0.844 | 4.47 | 7.37 | 2.2 |
| 20:40:40 | Wet | $(f_b, v_w, v_s, a_p)$ = (6, 12, 15, 0.1) | 143.945 | 7.77 | 0.870 (tied) | 146.954 | 7.226 | 0.864 | 2.08 | 7.23 | 0.69 |
| | | $(f_b, v_w, v_s, a_p)$ = (6, 12, 25, 0.1) | 150.119 | 6.32 | | 154.771 | 6.69 | 0.867 | 3.099 | 5.85 | 0.35 |
| 15:25:60 | Dry | $(f_b, v_w, v_s, a_p)$ = (6, 12, 15, 0.08) | 204.565 | 1.01 | 0.860 | 195.43 | 1.09 | 0.852 | 4.46 | 7.9 | 0.93 |
| 100:0:0 (temperature only) | Wet | $(f_b, v_w, v_s, a_p)$ = (3, 12, 15, 0.07) | 109.785 | 12.24 | 0.991 | 111.69 | 12.25 | 0.989 | 1.73 | 0.16 | 0.20 |
| 0:0:100 (total cost only) | Dry | $(f_b, v_w, v_s, a_p)$ = (6, 12, 15, 0.07) | 191.799 | 1.08 | 0.995 | 193.33 | 1.08 | 0.994 | 0.80 | 0 | 0.10 |

## 4. Conclusions

In this paper, the surface grinding process of AISI 4140 steel under two conditions-dry and wet have was optimized by using a combined methodology based on response surface methodology (RSM) and desirability functional approach (DFA). The combined methodology of RSM and DFA in optimization of machining parameter is rarely found in this relevant field and it is efficient in determining the optimum antagonist responses (here Temperature and *MRR*) which are led by different input machining parameters. Mathematical models were developed for contact temperature, *MRR*, and cost under both dry and wet conditions in order to correlate with the four process parameters: wheel velocity, workpiece velocity, depth of cut, and cross-feed rate. The model for *MRR* was derived analytically using Equation (1). Empirical models for temperature and cost were developed using RSM. It is verified that the experiments were conducted in the right way and uncertainty in input parameters was lower. The mean error for temperature and the total cost were less than 2% and 5%, respectively. In this study, we assumed that there was a negligible amount of heat transfer due to convection and radiation. In addition, for the total cost calculation, cost for the depreciation of grinding wheel was ignored, only utility cost, cost of cutting fluid, and direct labor cost were considered. The results suggest that:

1.  The wet condition dominates the overall desirability when the temperature is given the highest (≥20%) weightage. As our optimization goal was minimum temperature, the obtained result is as expected. Since the wet conditions keep the temperature down, the obtained result is achieved so. Again, in current scenario, cross-feed, wheel velocity, and depth of cut were at the lowest levels as a higher value of these parameter increases the amount of material removal per pass leading to increased temperature.

2.  Dry condition achieves the highest overall desirability if the cost is given the highest priority instead. As the optimization goal was minimum cost, the obtained result is as expected. The underlying reason of the obtained result is no use of cutting fluid in the dry condition, hence cost is reduced. However, in the wet condition, cost is added due to cost of cutting fluid and extra power consumption. Again, in the current scenario, except depth of cut and wheel velocity, all other input parameters are at their upper limit. This provides a higher material removal leading to short machining time and reduced cost. However, depth of cut and wheel velocity at its upper limit provides

larger power consumption which in turn increases cost. Thus, depth of cut and wheel velocity was kept at the lowest level.

3.     If *MRR* is given priority (20:40:40), i.e., quick material removal, then all the input parameters are set at their highest level along with the wet machining condition. Larger *MRR* require higher cross-feed, depth of cut, wheel velocity, and higher workpiece velocity. Due to higher *MRR*, quick removal of chips is required which is very effective with wet machining condition. The obtained result exists at two levels of wheel velocity (15 and 25 m/s). At higher wheel velocity, grinding zone temperature is higher, which is expected. However, it may lead to poor surface quality. Hence, lower wheel velocity is found to be more suitable for grinding.

4.     For equal weights of responses, the optimal values are found to be 6 mm/pass of cross-feed, 12 m/min of workpiece velocity, 15 m/s of the wheel or cutting velocity, 0.095 mm of the depth of cut in wet condition, with a maximum overall desirability value of 0.863. Optimum value for other weights of responses can be found in Table 8.

5.     For dry cutting condition, keeping other parameters constant, desirability value increases with higher values of depth of cut until 0.095 mm, from where it decreases. This is because, up to a certain increase in depth of cut, high temperature is produced which drives the overall desirability down. If depth of cut is kept constant instead, then a positive relationship between the overall desirability values, and cross-feed and workpiece velocity is observed.

6.     For the wet conditions, it is observed that workpiece velocity and cross-feed are positively related to the higher overall desirability values whereas the depth of cut and wheel velocity have negligible effects.

7.     For a specific weight assignment, it is evident that the depth of cut positively affects the overall desirability values in the dry cutting condition, but it has a negligible effect in the wet condition.

8.     It must be kept in mind that these conjectures are not invariable; rather it depends on how the weights are assigned. Wheel velocity has no apparent effect on desirability when all weights are equal in our case. However, this scenario will most likely change if different weights are assigned.

9.     The confirmatory experiments corroborate that the predicted responses are in good agreement with the experimental results.

The presented study is helpful in setting the input parameters when any of the output responses is given more priority compared to rest. Most important output responses are given higher weightage than others to determine the corresponding input parameters while calculating the overall desirability. This model is highly flexible for the researchers and industrial practitioners to set input parameters if a particular response is given more focus. Again, as the results indicate, for higher *MRR* all the input parameter, i.e., cross-feed, wheel velocity, workpiece velocity, and depth of cut were set at the highest level for quick material removal where surface condition is not an issue of concern. This particular scenario is used in rough machining (manufacturing low category engineering products) for shortest machining time at the light engineering workshops in developing countries i.e., Bangladesh. Again, if surface condition (low grinding zone temperature) is given priority (manufacturing better quality product) weightage scheme 100:0:0 can be used. This model can be adopted by the light engineering workshops in developing countries, i.e., Bangladesh. Another interesting future research endeavor might include efforts to see how multi-response optimization of machining parameters turns out if different prediction models are used instead of RSM. Especially, a machine learning model might be a better predictor than RSM, given sufficient data for training.

## 5. Strength and Limitations of the Study

*MRR* and cost was calculated based on direct measurement, so these output responses were accurate at its best. However, the temperature was measured based on the developed thermocouple setup presented in the study. This set up has some strengths i.e., simple,

rugged, easy to build the setup and relatively inexpensive. Further it can measure wide range of temperature and thermocouple junction can easily be inserted into the body cavity of the workpiece. So, the obtained temperature was best achieved as it was in direct contact with workpiece.

Besides, the developed setup has some limitations, i.e., limited accuracy and sensitivity, and is not convenient for practical applications. As mentioned at the experimental setup, the inserted pole into the workpiece cavity was insulated to avoid convection and radiation heat transfer, but it could not be fully nullified. Again, low voltage signals are susceptible to noise. As no shielding was there to stop noise, the results may have some errors.

**Supplementary Materials:** The following supporting information can be downloaded at: https://www.mdpi.com/article/10.3390/coatings12010104/s1, Table S1: Specifications of the workpiece and wheel used in the experiment; Table S2: Input parameters, their level and output units of each parameter; Table S3: Formula for calculating wheel velocity and *MRR*; Table S4: Raw data for the regression analysis under dry condition; Table S5: Raw data for the regression analysis under wet condition; Table S6: Data table showing design of experiment; Table S7: Data table showing the calculation of temperature using RSM equations; Table S8: Data table showing cost calculation under dry and wet condition using RSM equations; Table S9: Data table showing the calculation of desirability at dry condition; Table S10: Data table showing the calculation of desirability at wet condition.

**Author Contributions:** Conceptualization, R.R., S.K.G., T.I.K. and M.M.R.; methodology, R.R., S.K.G., T.I.K., S.H., M.M.R. and M.K.; software, S.K.G. and T.I.K.; validation, R.R., M.M.R. and T.I.K.; formal analysis, R.R., S.K.G., T.I.K. and M.M.R.; investigation, R.R. and T.I.K.; resources, R.R., M.K. and M.M.R.; data curation, R.R.; writing—original draft preparation, R.R.; writing—review and editing, M.M.R., S.K.G., T.I.K., T.A. and M.K.; visualization, S.H.; supervision, M.M.R.; project administration, M.M.R., M.A. and M.K.; funding acquisition, M.A. and M.K. All authors have read and agreed to the published version of the manuscript.

**Funding:** This work was supported by the National Research Foundation of Korea (NRF), grant funded by the Korea government (MSIT) (No. NRF-2019R1G1A1099335).

**Institutional Review Board Statement:** Not applicable.

**Informed Consent Statement:** Not applicable.

**Data Availability Statement:** Data sharing is not applicable to this article.

**Acknowledgments:** The authors are grateful to the Department of Industrial and Production Engineering, BUTEX for permitting to employ the lab facilities to conduct all the necessary experiments.

**Conflicts of Interest:** The authors declare no conflict of interest.

## Nomenclature

| | |
|---|---|
| $a_p$ | depth of cut (mm) |
| $b$ | width of the cut (mm) |
| $b_0$, $b_i$, $b_{ii}$, $b_{ij}$ | coefficient of intercept, linear, quadratic and interaction of input variables respectively |
| $b_{jo}$, $b_{jk}$, $b_{jkk}$, $b_{jkl}$ | are the regression coefficients |
| $Cost_{dry}$ | total cost at dry machining condition (BDT) |
| $Cost_{wet}$ | total cost at wet machining condition (BDT) |
| $D_i$ | overall desirability of all responses |
| $a_{pi}$ | the depth of cut at experiment i |
| $d_{ij}\left(\hat{Y}_{ij}\right) = 0$ | denotes the lowest desirability value of output response (*Yij*) |
| $d_{ij}\left(\hat{Y}_{ij}\right) = 1$ | denotes the highest desirability value of output response (*Yij*) |
| $f_b$ | cross-feed (mm/pass) |
| $f_{bi}$ | cross-feed at experiment *i* |
| $G_{ij}$ | target value of the jth response (*Yij*) $0 \le d_{ij}\left(\hat{Y}_{ij}\right) \le 1$ |
| $G_{ij}^{max}$ | maximum goal of output *j* at experiment *i* |

| $G_{ij}^{min}$ | minimum goal of output $j$ at experiment $i$ |
| --- | --- |
| $m$ | no. of experimental runs |
| $M_{ij}$ | lower level of output $j$ at experiment $i$ |
| $MRR$ | material removal rate ($mm^3/s$) |
| $n$ | no. of output responses variables |
| $N_{ij}$ | higher level of output $j$ at experiment $i$ |
| $p$ | no. of input independent parameters |
| $Temp_{dry}$ | Temperature at dry machining condition ($°C$) |
| $Temp_{wet}$ | Temperature at wet machining condition ($°C$) |
| $v_s$ | wheel velocity (m/s) |
| $v_{si}$ | the wheel velocity at experiment $i$ |
| $v_w$ | workpiece velocity or longitudinal table travel velocity (m/min) |
| $v_{wi}$ | workpiece velocity at experiment $i$ |
| $Y_{ij}$ | output responses value at ith experiment of jth response, $j = 1, 2, \ldots, n$ |

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
