# Peer review of "Multi-Response Optimization of Surface Grinding Process Parameters of AISI 4140 Alloy Steel Using Response Surface Methodology and Desirability Function under Dry and Wet Conditions"

_coatings, doi:10.3390/coatings12010104_

Round 1

Reviewer 1 Report

Journal:  Coatings

Ref manuscript ID coatings-1509973

Manuscript titleMulti-response optimization of surface grinding process parameters of AISI 4140 alloy steel using response surface methodology and desirability function under dry and wet conditions”

Comments to authors:

Although the manuscript entitled " Multi-response optimization of surface grinding process parameters of AISI 4140 alloy steel using response surface methodology and desirability function under dry and wet conditions " is well written.

However, the minor revision is suggested in order to get the article published in the journal. The author needs to incorporate the following suggestions.

  • the entire document seriously includes countless spelling and grammar errors in several places which must be thoroughly polished and refined by the authors before acceptance
  • Improve the physical illustrations of the problem.
  • Provide the appropriate references of Eq. (1)- (2) and Eq. (3)-(11).
  • The authors should explain more about the novelty of their work which is not clear in introduction, the work objective it is not clearly written.
  • In Response Surface Methodology, based on the responses acquired in the experiments, Regression Analysis is utilized to identify the relationships between the responses and the variables to establish a mathematical model that satisfies the relationship between a group of test factors and objective functions. This model is then used to explore the optimal solution in the experimental area. the authors must clarify the relevant approach to the problem by justifying the choice of the method used and what is the goal of the optimization
  • Add the following reference.
  • Thermo-Mechanical Coupled Analysis-Based Design of Ventilated Brake Disc Using Genetic Algorithm and Particle Swarm Optimization, SAE Int. J. Passeng. Cars - Mech. Syst.14(2):137-150, 2021
  • In 'Result and Discussion' authors have noted observations. But it is suggested that to provide physical explanations of all obtained results which can enrich the quality of the paper.

Altogether, the paper needs modification to be suitable for the standards required for publication; therefore I recommend that it required to “Minor revision”.

I look forward to receiving the revised version of this manuscript

My best regards

Reviewer 2 Report

Reviewed article concerns multi-response optimization of surface grinding process parameters of AISI 4140 alloy steel using response surface methodology and desirability function under dry and wet conditions and is write in accordance with generally accepted standards of the scientific works. However, after a careful reading of it, I find that the results obtained by the authors contradict the knowledge known and provided many times in the literature concerning the influence of grinding parameters on its results.

In the adopted methodology, grinding speeds were used at least 2-3 times lower than those commonly used in this type of processes (15-25 m/s were assumed, while grinding at 60-100 m/s is commonly used). Moreover, as a result of extensive analyses, the authors indicate as the most beneficial grinding at a speed of 15 m/s, which completely contradicts the known influence of the grinding speed on the basic phenomena in the grinding zone. For decades, efforts have been made to increase the grinding speed in order to obtain more favourable thermal conditions (removal of heat along with the chip), lower roughness of the machined surface (a greater number of contacts between grinding wheel and workpiece surface results in smaller cross-sections of layers cut with a single cutting vertices) and reduction of grinding wheel wear (extension of tool life). Of course, there are exceptions to this rule and these apply to hard-to-cut materials. However, the steel being machined is not one of them.

The results obtained by the authors challenge the basic knowledge of abrasive machining and, as such, may be misleading for the readers. For this reason, I am against the publication of the submitted article.

Reviewer 3 Report

The reviewer comments of the paper «Multi-response optimization of surface grinding process parameters of AISI 4140 alloy steel using response surface methodology and desirability function under dry and wet conditions»- Reviewer

The authors presented an article «Multi-response optimization of surface grinding process parameters of AISI 4140 alloy steel using response surface methodology and desirability function under dry and wet conditions». However, there are several points in the article that require further explanation.

Comment 1:

The introduction needs to be improved.

Now there are many references published more than 5 years ago, so the list of references needs to be supplemented with at least 4-5 more references published over the past 5 years. Here are some recent articles:

International Journal of Advanced Manufacturing Technology 2021, 117(3-4), 729–750. doi:10.1007/s00170-021-07785-x

Metals 2020, 10(7), 895. doi:10.3390/met10070895

After analyzing the literature, show before formulating the goal of the "blank" spots. Which has not been previously done by other researchers. You must show the importance of the research being undertaken. Show what will be the new research approach in this article. You need to show a hypothesis.

Comment 2:

  1. Methodology &Experimental Procedures

It would be better for Depth of cut to use ap instead of d.

Are all figures original? If not needed appropriate citations and permissions. Refine this for figures throughout the article.

Add the details of the grinding wheel parameters: grit, bond, etc. Explain your vow.

Describe the measurement procedure in more detail. At what point in time? How is the measuring setup set up? How many repetitions of measurements? What statistical methods are used to process experimental results? Describe the experimental stand in more detail. What method of experiment planning is used and why?

Comment 3:

Conclusions.

It is necessary to more clearly show the novelty of the article and the advantages of the proposed method. Add qualitative and quantitative results of your work. What is the difference from previous work in this area? Show practical relevance.

The article is interesting, but needs to be improved. Authors should carefully study the comments and make improvements to the article step by step. After major changes can an article be considered for publication in the "Coatings".

Round 2

Reviewer 2 Report

Reviewed article concerns multi-response optimization of surface grinding process parameters of AISI 4140 alloy steel using response surface methodology and desirability function under dry and wet conditions and is write in accordance with generally accepted standards of the scientific works. After careful reading of the submitted text there are some substantive remarks that should be taken into consideration by the Authors to improve reviewed text.

  1. Presented study widely covers defined scientific problem and with experimental investigations provides proper background for given conclusions, however deeper scientific consideration of obtained results referred to the basic phenomena in grinding processes should be given.
  2. I suggest also to give wider description of potential use of presented findings in scientific research as well as in industrial practice.
  3. In the discussion section should be provide more references to already known results from literature.
  4. The strengths and limitations of the obtained results and applied methods should be clearly described.
  5. In conclusions deeper explanation of observed phenomena and trends should be given (conclusions should refer not only to results but also to causes of obtained results).

Reviewer 3 Report

In table 9, replace d with ap. And in figure 4. And in formula 1. Check all the text again.
Page 3 contains references that are not found in the main file. Authors should check everything carefully.

Author Response

This manuscript is a resubmission of an earlier submission. The following is a list of the peer review reports and author responses from that submission.

Round 1

Reviewer 1 Report

In this paper, the authors studied the relation between the input variables (cross-feed, table velocity, wheel or cutting velocity, depth of cut) and responses (temperature, MRR, cost) under wet and dry conditions of the surface grinding process. But the way of presenting the research discovery is not clear for readers to understand the authors’ ideas and findings, plus there were also significant elements missing in the information flow. Please see detailed comments below:

  1. Very critical information in several aspects are missing:
    1. Line 279, how DFA approach was conducted? Where is figure 4
    2. Line 314-329, only temperature and cost were discussed, where is MRR?
    3. Line 314-329, how these models/equations were obtained? Where were the raw data for regression analysis?
    4. Line 381-384, what are equation 4,5,6,7. There were no indication about these equations
    5. Design of experiment is missing for raw data collection, which is the fundamental information for optimization based manuscript
    6. How table 6 was obtained is unclear
    7. The influence of input variables (cross-feed, table velocity, wheel or cutting velocity, depth of cut) on the RSM plot was barely discussed in figure 8 and 9
    8. Parameter settings claimed to be optimal are better to be validated
  2. Page 1, abstract: The current abstract didn’t display all the aspect of discoveries, for example, optimal settings for dry condition was missing. Some content should better be discussed in methodology section, such as line 25-26.
  3. Instruction in the template should be deleted. For example, line 146-152; line 294-296
  4. Duplicated information. For example, line 284-291
  5. Figures resolution is low and blur
  6. Line 194, abbreviation of EMF should be explained
  7. Some minor issues about unit: line 172, superscript; table 5, unit missing

Author Response

kindly find the reply.

Reviewer 2 Report

Below are listed some substantive remarks that should be taken into consideration by the Authors to improve reviewed text:

  1. the abstract should include information about new methods, results, concepts, and conclusions – in its current form, the abstract needs to be rewritten to include more information on motivation and research gap,
  2. the novelty of given approach should be emphasized in introduction,
  3. the first paragraph in section 2 is from guidelines for authors and should be removed,
  4. I suggest providing more precise information about used experimental (producer of the grinding machine) and measurement positions (types and producers),
  5. grinding is a process to finish surface (low surface roughness without temperature defects of workpiece structure) and obtain proper shape and dimension accuracy. In presented study Authors did not mention about accuracy of grinding or surface texture (or surface integrity) after machining using two different conditions (wet and dry). I strongly advise to add results concerning this issue,
  6. some of figures are very poor quality (unreadable) – please try to provide graphics with better resolution,
  7. parameter “table velocity” is named by standard as “table feed velocity vf” and it could be for example axial table feed velocity vfa or radial table feed velocity vfr according to direction refereing to grinding wheel – please name precise grinding parameters according to international standard (for example ISO standard),
  8. use proper symbols of parameters (also on figures) – capital V is for volume not velocity (symbols should be writing according to writing in standard),
  9. vw is a symbol for workpiece velocity not “wheel velocity” – please study basic terminology in grinding,
  10. vt -s not a proper symbol for feed velocity – please study basic terminology in grinding,
  11. Authors use different designation for the same parameters (Vw and WV, Vt and TV) – pleas use standarized symbols and names of grinding parameters,
  12. what was the grinding parameters?
  13. what was the WET grinding conditions (coolant flow rate, was used oil – Castrol Syntilo or emulsion made on its basis, what was the proportion ratio for emulsion, what was the geometry/type of nozzle, how many nozzles were used)?
  14. what wat the dressing cut parameters?
  15. what was the number of repetitions of experiments?
  16. symbols used in equitation for MRR are not correct (precise) according to commonly used standards in grinding technology,
  17. all used symbols should be re-check and correct if necessary, according to basic terminology in grinding,
  18. in the discussion section should be provide more references to already known results from literature,
  19. the strengths and limitations of the obtained results and applied methods should be clearly described,
  20. in conclusions deeper explanation of observed phenomena and trends should be given (conclusions should refer not only to results but also to causes of obtained results),
  21. the conclusions should highlight the novelty and contribution to the state of the knowledge in given area.

Reviewer 3 Report

Manuscript Number : Coatings-1297749

Coatings

<Manuscript Title> Multi-response optimization of surface grinding process parameters of AISI 4140 alloy steel using response surface methodology and desirability function under dry and wet conditions

 Reviewers' comments:

After reading the paper, I rate this paper an interesting work, which addresses important problems. The results in the paper tend to be right, and the method is possible to give better results. Judging from what I rate, I recommend Coatings  to accept the paper but after doing a major revision.

 Please, during the revised version, put the requested corrections and changes with a distinct red or blue color that allows the changes to be tracked in the text for easier review

  • Add the list of symbols as nomenclature before the introduction section? since the document contains several incomprehensible symbols and characters which are inserted in several places in the text and in the set of equations. In addition, there are several unclear and incomprehensible abbreviations in the text that need to be clarified in this nomenclature
  • The document contains ample information repeated in several places and should be revised carefully, eliminating double reporting
  • Many places typographical errors are there. Authors should take care typographical

errors in the entire manuscript. therefore, English must be polished and carefully refined.

  • The introduction ends by stating the goals and objectives of the project. The introduction ends by stating the goals and objectives of the project. In this part, authors should clearly and fully describe the objective of their study.
  • The style of references in the document is poorly written and requires revision according to journal guidelines
  • what is the name of the software used in the contour plot ??
  • The document contains in the theoretical part several equations which have been inserted without any source or bibliographic references, so please provide these references according to the journal format.
  • The introduction section is adequate but needs some language revision.
  • Authors should explain more about the novelty of their work in introduction
  • The authors presented the results, but they didn't give a sufficient discussion for the results. This makes the paper look like a lab report rather than a research paper.
  • From the introduction, it seems that the authors know little about what has been done by others in this field. A detailed literature review is strongly recommended. 
  • Key assumptions and their implications could have been elaborated
  • The authors should think over the real significance of their results and try to rewrite this section to improve understanding of the conclusions
  • The quality of the figures in this document needs to be improved; the figures need to be larger in size so the data and labels can be clearly read.
  • The Introduction section is meant to set the context for your research work and highlight how it contributes to the knowledge in your field and builds on previous similar studies. The introduction should provide a clear statement of the problem, the relevant literature on the subject, and the proposed approach or solution. It is be understandable to colleagues from a broad range of scientific disciplines. For that purpose I would like request author to add the following recent papers, which can help to enhance the introduction section:

1)      Computational fluid dynamics (CFD) analysis and numerical aerodynamic investigations of automotive disc brake rotor, Australian Journal of Mechanical Engineering  16 (3), 2018  , pp 188-205

  • - Numerical study of heat convective mass transfer in a fully developed laminar flow with constant wall temperature Case Studies In Thermal Engineering, 6 , 2015,  116-12
  • - Analytical solution and numerical simulation of the generalized Levèque equation to predict the thermal boundary layer, Mathematics and Computers in Simulation. 180 , pp.43–60 (2021)
  • - Numerical study of heat transfer in fully developed laminar flow inside a circular tube. J. Adv. Manuf. Technol.85 (9), 2681–2692 (2016)
  • - An analytical method for solving exact solutions of the convective heat transfer in fully developed laminar flow through a circular tube, Heat Transfer Asian Research, 2017, 46(8),pp 1342-1353
  • - Numerical simulation of thermally developing turbulent flow through a cylindrical tube, J. Adv. Manuf. Technol.102 (5-8), 2001–2012 (2019)
  • The authors must provide a greater discussion of the results.
  • The conclusions should be re-written, The authors should think over the real significance of their results and try to rewrite this section to improve understanding of the conclusions. The authors must at the end of the conclusions draw a series of adopted perspectives and recommendations for this study.

Overall, this manuscript requires major changes to make it worth publishing in Coatings I look forward to receiving the revised version of this manuscript

Best Regards

Round 2

Reviewer 1 Report

Please see detailed comments below for the revised manuscript and responses:

  1. Regarding original comment 1.1.: Figure 4 quality needs to be improved, currently it looks like two figures overlay on each other. Also, some text in the box was blocked.
  2. Regarding original comment 1.3.: Still cannot see the raw dataset.
  3. Regarding original comment 1.5.: Still cannot see the 80 runs for DOE. In addition, the line numbers provided by the authors in the manuscript are useless. Author stated “applied to run 80 experiments for this research work which is delineated in line 195 – 198 in the revised manuscript.”, but the actual line number is 260-262.
  4. Regarding original comment 1.7.: Cannot find the discussion added in revised manuscript in line 373-399 and in line 483-528
  5. Regarding original comment 1.8.: Hypothesis needs validation to be scientifically sound. Previous literature is other researchers’ work…

Reviewer 2 Report

Authors take into considration all of my comments.

Article can be published in revised version.

Reviewer 3 Report

Journal:  Coatings

Ref manuscript ID coatings-1297749

Manuscript title Multi-response optimization of surface grinding process parameters of AISI 4140 alloy steel using response surface methodology and desirability function under dry and wet conditions

Assistant Editor

I have completed the review process of the revised version of the above-mentioned document, and after careful reading and focused on it, and after careful checking of all the changes inserted in the document, accuracy of the figures, tables ,equations ,  and citation references, my final comments can be found below:

Comments:

The authors have satisfactorily responded to all my questions and made the necessary changes to the manuscript.

Recommendation: Accept With No Changes

Thank you again for giving me a chance and inviting me to review this document.

Best regards
